



# Quantifying paleo-reconstruction skill of the Southern Annular Mode in a model framework

Willem Huiskamp[1,2] and Shayne McGregor[3]

[1]RD1- Earth System Analysis, Potsdam-Institute for Climate Impact Research (PIK), Member of the Leibniz Association, Potsdam, Brandenburg, Germany
[2]Climate Change Research Centre, UNSW Sydney, Sydney, NSW, Australia
[3]School of Earth Atmosphere and Environment, Monash University, Melbourne, Victoria, Australia

**Correspondence:** Willem Huiskamp (huiskamp@pik-potsdam.de)

**Abstract.** Past attempts to reconstruct the Southern Annular Mode (SAM) using paleo archives have resulted in records which can differ significantly from one another prior to the window over which the proxies are calibrated. This study attempts to quantify not only the skill with which we may expect to reconstruct the SAM, but also assess the contribution of regional bias in proxy selection and the impact of non-stationary proxy-SAM teleconnections on a resulting reconstruction. This is achieved using a pseudo-proxy framework with output from the CM2.1 global climate model. Reconstructions derived from precipitation fields perform better, with 89% of reconstructions calibrated over a 61 year window able to reproduce at least 50% of inter-annual variance in the SAM, as opposed to just 25% for surface temperature (SAT) derived reconstructions. Non-stationarity of proxy-SAM teleconnections, as defined here, plays a negligible role in reconstructions, however the range in reconstruction skill is not negligible. Reconstructions were most likely to be skilful when proxies are sourced from a geographically broad region, with a network size of 70 proxies.

## 1 Introduction

The Southern Annular Mode (SAM) is the leading mode of atmospheric variability in the Southern Hemisphere which nominally describes the intensity and latitudinal location of the subtropical westerly jet. Positive and negative phases of the SAM have been linked to changes in surface air temperature and precipitation (Figure 1b and d respectively) in Australia (Hendon et al., 2007), New Zealand (Gallant et al., 2013) as well as South America and Africa (Gillet et al., 2006; Silvestri and Vera, 2009). For example, positive phases of the SAM over the period 1979-2005 are typically associated with negative annual temperature anomalies over the Antarctic continent (Thompson and Solomon, 2002; Kwok and Comiso, 2002; Gillet et al., 2006) and positive anomalies over the Antarctic Peninsula, southern South America and southern New Zealand (Kwok and Comiso, 2002; Silvestri and Vera, 2009). Precipitation changes typically found during a positive SAM phase include negative annual precipitation anomalies over southern South America, New Zealand and Tasmania and positive precipitation anomalies over



Australia and South Africa (Silvestri and Vera, 2009). The SAM also has a measurable impact on changes in ocean carbon fluxes (Lovenduski et al., 2007; Lenton and Matear, 2007; Le Quéré et al., 2007; Huiskamp and Meissner, 2012; Hauck et al., 2013; Huiskamp et al., 2015; Keppler and Landschützer, 2019) and heat uptake (Russell et al., 2006; Liu et al., 2018).

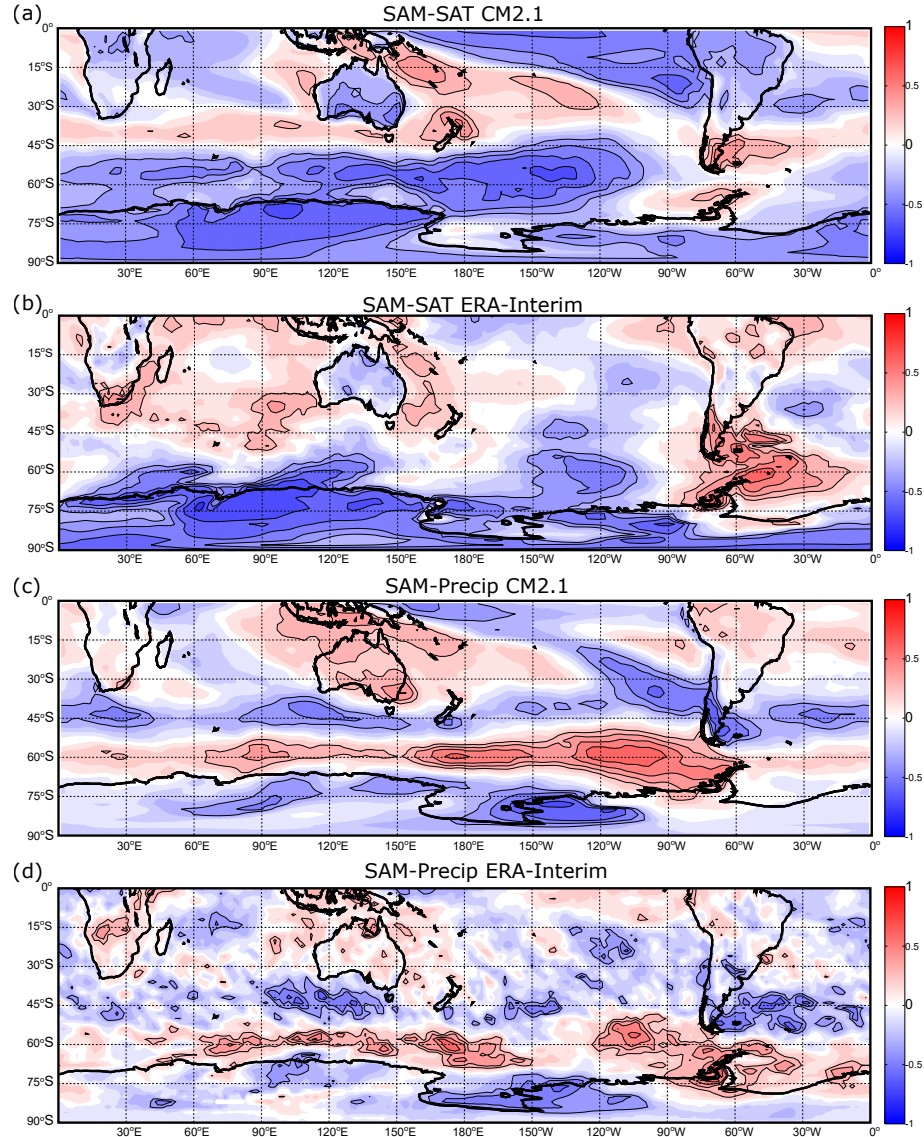

**Figure 1.** Correlations of annual-mean (Jan-Dec) SAT (a) and precipitation (c) from the GFDL CM2.1 model with the model-derived SAM, calculated over 500 years. Panels b) and d) show annual-mean correlations between the ERA-Interim (Dee et al., 2011) reanalysis product (SAT - b, Precip - d) with the Marshall SAM index (Marshall, 2003), calculated over a 36 year period from 1979-2014. Contours show $r \geq |0.3|$.



Over the last five decades the SAM has shown a trend towards more positive values, consistent with a poleward intensification of the surface westerly winds that has been largely attributed to anthropogenic forcing, such as stratospheric ozone depletion and the increase in atmospheric $CO_2$ (Son et al., 2008; Lee and Feldstein, 2013; Previdi and Polvani, 2014). In addition, both high frequency (3-4 months) and low frequency (16 years) variability has been observed in the SAM as derived from reanalysis experiments (Raphael and Holland, 2006). It is important to place these present day changes into context in order to understand

the contributions of forced and natural variability to these observed multi-decadal trends. Following on from this, this would allow us to examine the potential impacts of persistent, multi-decadal to centennial changes in the winds on regional weather patterns, large scale ocean circulation, and even the marine carbon cycle .

Instrumental reconstructions of the SAM extend as far back as 1865 (Jones et al., 2009) but involve significant uncertainties pre-mid $20^{th}$ Century due to fewer observations and the methods used to compensate for this (i.e, estimates based on atmo-

spheric conservation of mass, such as Jones et al. (2009)). Direct measurements, meanwhile, only extend as far back as 1958 (Marshall, 2003). If we wish to extend our understanding of SAM trends and variability back beyond the instrumental record, reconstructions derived from paleo archives are required.

## 1.1 Paleo reconstructions of SAM variance

These reconstructions can be made by examining changes preserved in natural environmental archives (biological, chemical

and physical records), which are considered to be sensitive to climatic impacts of the mode of variability we seek to reconstruct. In the case of the SAM, this has traditionally been achieved by finding proxies that are sensitive to both precipitation or surface air temperature changes associated with the two different phases of the SAM. For example, a positive SAM is associated with a poleward intensification of the westerlies, resulting in a southward migration of the storm tracks and drying in South America, New Zealand and Tasmania (Gillet et al., 2006), and cooling over much of Australia and Antarctica (Schneider et al., 2006).

Proxies recording changes in temperature and precipitation include tree ring records, ice cores and terrestrial sediment core records, although the latter are less favoured due to chronological uncertainties and a typically lower temporal resolution. Growth of trees, and therefore ring width, can be sensitive to both temperature and precipitation depending on the tree type and its location, while ice cores can provide accumulation rates, $\delta^{18}O$ and $\delta D$ (e.g. Steig et al. (2005)) - which record air temperature and precipitation (accumulation). Finally, coarse particle dust records provide a direct proxy of atmospheric circulation

(Koffman et al., 2014).

### 1.1.1 Reconstructions and their potential issues

The relationship to the SAM is typically established by correlating changes in these proxy records with a SAM index developed from instrumental or reanalysis data over a period spanning several decades (Villalba et al. (2012) and references therein; Abram et al. (2014)). Individual records are then combined into a single index using a reconstruction method such as

a regression approach (Villalba et al., 2012) or weighted Composite Plus Scaling (CPS)(Abram et al., 2014; Dätwyler et al., 2018).





There are, however, two fundamental assumptions being made when we reconstruct past climate in this way. Firstly, we assume that a hemisphere-wide mode of variability can be accurately reconstructed using records from a geographically limited sample space. As there is relatively little land in the Southern Hemisphere, particularly in the latitude of the westerlies, SAM

reconstructions often rely disproportionately on records from narrow longitude bands. Sites are primarily in South America, Australia and New Zealand (Villalba et al., 2012), and Antarctica, with Antarctica being the only location able to provide samples with good longitudinal coverage. Abram et al. (2014) suggest their regional, Drake Passage sector SAM reconstruction is representative of the hemispheric mean signal by extracting a sea level pressure-derived (SLP) SAM index from a suite of 8 global climate models and then compare it with a secondary SAM index that is derived from the same SLP field, but restricted to

the Drake Passage sector. They find that the regional expression of the SAM in these models closely resembles the hemispheric expression over 1000 years, a conclusion supported by the regional SAM records of Visbeck (2009). Dätwyler et al. (2018) on the other hand find non-trivial differences between their hemisphere-wide SAM reconstruction and that of Abram et al. (2014), implying that an annual-mean SAM reconstructed from paleodata is not well approximated by sampling from a limited region.

Secondly, when we correlate a proxy to modern SAM over a calibration window of several decades, we make the assumption

that this relationship remains the same through time. This is commonly referred to as proxy stationarity. Gallant et al. (2013) investigated this previously using instrumental data spanning the period 1900-2009 and reported that 21-37% of Australian precipitation records showed non-stationary teleconnections to ENSO and SAM. Silvestri and Vera (2009) performed a similar study with observed precipitation and surface air temperature (SAT) records from Australia and South America spanning the 1960-70's and 1980-90's spring months. They find significant positive correlation of the SAM with SAT in the Australia/ New

Zealand region in the earlier decades can become insignificant or even negative in the more recent decades. Dätwyler et al. (2018) built on this by adding a stationarity criteria to their proxies for reconstructing the SAM, but at a cost of calibrating their proxies with a longer, but less reliable record (Jones et al., 2009). The resulting reconstructions showed a more stable teleconnection through time, but were not necessarily more skillful (as measured by validation statistics). Finally, when considering a sufficiently short calibration period such as a decade or two, stochastic noise or another climate signal (eg. ENSO) can

modulate the correlation strength between, for example, South American precipitation and the SAM without the precipitation record being classified as non-stationary (Yun and Timmermann, 2018).

With this in mind, this study aims to quantify the uncertainties raised by the aforementioned assumptions within a modelling framework, similar to Batehup et al. (2015). This study will address the following questions: 1) What impact does proxy network size and calibration window size have on the skill of a resulting reconstruction? 2) How does the geographical range of the proxies affect reconstruction skill? 3) Are any regions in our model framework prone to producing non-stationary proxies

and what could be modulating the SAM-proxy teleconnection?



## 2 Methods

### 2.1 Model and data

The data used in this study is a 500-year pre-industrial control simulation of the GFDL CM2.1 climate model, with all boundary
conditions set to CE 1860 levels. This assures that any changes in the model SAM are due to internal variability in the climate
system only. The model is a fully-coupled global climate model with ocean (OM3.1), atmosphere (AM2.1), land (LM2.1) and
sea ice components (SIS). The ocean model has a resolution of $1°$x $1°$ which increases equatorward of $30°$ to a meridional
resolution of $1/3°$ at the equator (Griffies et al., 2005). The atmospheric and land surface models have a resolution of $2°$
latitudes by $2.5°$ longitude and the AM2.1 has 24 vertical levels (Delworth et al., 2006).

CM2.1 was selected due to its relatively good representation of the SAM. Figure 1 shows correlations of surface air tem-
perature (a) and precipitation (c) with the model-derived SAM index, which agree spatially well with the analysis of both the
ERA-40 (Karpechko et al., 2009) and ERA-Interim (Dee et al., 2011) reanalysis datasets (Figure 1b and d). The spatial struc-
ture of the SAM is well simulated, accurately capturing the centre of action over the Pacific, while being slightly too zonally
symmetric on the eastern half of the Southern Hemisphere (Raphael and Holland, 2006). Importantly for our purposes, CM2.1
accurately simulates the latitude at which the SAM transitions from its positive to its negative node over South America, which
many models of a similar age and computational complexity fail to achieve (Raphael and Holland, 2006). The amplitude of
the model SAM index is comparable with observations (Raphael and Holland, 2006), although its variability is larger than
observed (Karpechko et al., 2009). The performance of CM2.1 when considering the response of SAT and precipitation to the
SAM is reasonable in comparison to other similar models (Karpechko et al. (2009), their Figure 3). Gallant et al. (2013) also
use CM2.1 in their study, citing its skill in simulating the teleconnection of the SAM with precipitation over Austral Winter.
Regional biases in SAT and location of the jet in this model are discussed further in Section 4.

The use of climate models to assess the skill of paleo reconstructions provides an opportunity to investigate a 'perfect'
time-series of the climate index we wish to reconstruct and the ability to reconstruct this index with fields from a model,
which act as pseudo paleo-proxies. As mentioned previously, these 'perfect' proxies are free from non-climatic noise that may
also degenerate a teleconnection signal in a real proxy, isolating instead changes in teleconnection strength due to underlying
variability in the climate. This is in contrast to 'real world' proxies which are also prone to other influences inherent with
the physical/chemical/biological nature of the proxy itself. For example, while we may use precipitation as an analogue for
tree ring records, these records in the real world are also sensitive to other factors such as non-linear growth trends, local
hydroclimate and temperature, soil characteristics and access to sunlight, to name a few (Fritts, 1976). It is often assumed that
these effects will be minimised by sampling proxies from a range of regions as these other factors would not be expected to be
correlated amongst differing locations. Using a model framework it is possible to robustly assess multi-decadal to centennial
changes in proxy teleconnection and how calibration over certain windows in time affects the skill of index reconstruction.

Paleo-proxies are not uniformly sensitive to one season or variable, depending on the region from which they are sourced.
For example, a tree ring record constructed by Cullen and Grierson (2009) from south-west Western Australia is shown to be
particularly sensitive to austral autumn-winter rainfall. Alternatively, South American tree ring records compiled by Villalba




et al. (2012) show sensitivity to summer-autumn precipitation, while New Zealand records appear to be most responsive to summer temperature. Finally, Koffman et al. (2014) present a coarse dust particle record which they demonstrate is highly correlated with 850hPa winds, particularly in summer when the jet migrates southward. In addition to this, while the proxies are sensitive to SAT or precipitation during one season, the SAM's strongest influence on these variables may be during a

different season entirely. For example, while the South American tree rings of Villalba et al. (2012) are sensitive to summer-autumn precipitation, the SAM signal is most clearly seen in late spring and winter precipitation in south-eastern South America (Silvestri and Vera, 2009). For the sake of simplicity and consistency, we simply employ annual mean (Jan-Dec) fields for sea level pressure, surface air temperature and precipitation and focus instead on the impact of network size and calibration window length rather than seasonal effects. The SAM index in the model is calculated according to the method of Gallant et al. (2013)

as the difference in normalized, zonally averaged sea level pressure anomalies between $40°$S and $60°$S.

## 2.2    Calculation of non-stationarity

A proxy is considered non-stationary when its teleconnection to SAM is changed by some dynamical process rather than stochastic variability (localised weather) such that the signal it records is no longer representing changes in the SAM. The teleconnection is established via a running correlation between the proxy and the SAM index over a window of 31, 61 or 91

years. We define non-stationary teleconnections following the method of Gallant et al. (2013) and Batehup et al. (2015). A proxy is considered nonstationary when variability in its running correlation with SAM exceeds what would be expected if the proxy were only influenced by local random noise.

Following from this, a Monte Carlo approach (van Oldenborgh and Burgers, 2005; Sterl et al., 2007; Gallant et al., 2013) is used to create stochastic simulations of SAT and precipitation at each grid point in the model. These stochastic simulations

were created to have the same statistical properties as the original SAT and precipitation data from the CM2.1 simulation. To determine the range of variability expected due to the stochastic processes mentioned previously, one thousand of these time series were created at each grid point according to the following equation from Gallant et al. (2013).

$$v(t) = a_0 + a_1 c(t) + \sigma_v \sqrt{1 - r^2}[\eta_v(t) + \beta\eta_v(t-1)] \tag{1}$$

$v(t)$ is the stochastic SAT or precipitation time series. $a_0 + a_1$ are regression coefficients representing the stationary tele-

connection strength between SAT or precipitation and the SAM index ($c(t)$) while the remaining terms represent the noise added to the time-series. A red noise, $[\eta_v(t) + \beta\eta_v(t-1)]$, is added and weighted by the standard deviation $\sigma_v$ of the local SAT or precipitation time-series as well as the proportion of the variance not related to the regression ($\sqrt{1-r^2}$) - where $r$ is the correlation between the SAT/precipitation time-series and SAM index. The red noise is a combination of random Gaussian noise $\eta_v(t)$ and autocorrelation of the SAT or precipitation time-series at a lag of one year ($\beta$).

The stochastic simulations of SAT and precipitation were used to create a 95% confidence interval for each grid point of all possible running correlations a time-series could have and still be considered to have a stationary teleconnection with SAM. Therefore, if the time-series from the CM2.1 simulation had a running correlation that fell outside the confidence interval, we





consider the grid point to be non-stationary with SAM in that temporal window, as it is unlikely to be affected by stochastic processes alone. It should be noted that as a 95% confidence interval is used, non-stationarity will be falsely identified 5% of the time, hence we define a grid point as non-stationary only if the running correlation falls out of the confidence interval more than 10% of the time, or 50 of the 500 years; more than double the 5% we might expect by chance alone. As correlations are bounded between ±1, the running correlations are converted to Fisher Z-scores:

$$Z = \frac{1}{2}\ln\left(\frac{1+r}{1-r}\right),\qquad(2)$$

where $r$ is the running correlation value.

## 2.3 Generation of pseudoproxies

SAT and precipitation fields from the model are used to represent climate proxies in the model, as discussed in Section 2.1. Rather than being inferred via changes in tree ring growth, these proxies are direct measures of these variables and therefore free of non-climatic noise (von Storch et al., 2009). We do not add noise to increase the realism of these proxies, rather we assess reconstruction skill and non-stationarity in a 'best-case-scenario' where we assume the proxy is a perfect analogue for the metric it is deemed to represent (SAT or precipitation), similar to the experiments of Dätwyler et al. (2020).

Proxies are randomly selected in accordance with several conditions. The proxy must be on land in the Southern Hemisphere and must have a correlation with the model SAM index of |0.3| or greater within the calibration window. The correlation of 0.3 is arbitrary in choice but ensures that the proxy represents the SAM to some extent while not being so high that proxies are only sourced from a geographically limited region. For this reason we rely on the reconstruction method to extract a clear SAM signal from these time-series.

Ideally, proxies would be calibrated with the SAM over the full length of the time-series - 500 years, however as previously noted, real world proxies are calibrated over shorter windows of several decades. For each gridpoint in the model, the time-series is split into 10 windows of either 31, 61 or 91 years in length, whose midpoints are evenly spaced throughout the 500 years, regardless of overlap or space between them. The number of proxies meeting our aforementioned criteria in each region/ windowsize can be found in Table 1. This approach allows for the possibility that a proxy may have a strong correlation with SAM over the calibration window, but which may be insignificant or even reversed over other windows, or indeed the full 500 years. These window sizes are chosen to assess the effect differing window lengths have on the resulting skill of the reconstruction. For example, the use of a 61 year calibration window, as opposed to a 31 year one, may decrease the effect of decadal climate variability and its modulation of the pseudoproxy-SAM teleconnection.

Reconstructions are computed with a network size of between 2 and 70 proxy records, a range typical of past reconstructions with strict selection criteria (eg. Abram et al. (2014) and Dätwyler et al. (2018)). This is done to quantify the dependence of reconstruction skill on network size. 1000 networks are generated for each of the 10 calibration windows for network size. Each site in each network is randomly selected and unique, while the same site may be included in more than one network.



To reconstruct the proxy networks into a single proxy-SAM index, we use the weighted composite plus scale (CPS) method
(Esper et al., 2005; Hegerl et al., 2007), similar to that used by Abram et al. (2014). As the scope of this study does not include
the effect of different reconstruction methods on the skill of the reconstructed index, CPS is used as it is commonly employed
in paleo-reconstructions (PAGES 2k Consortium, 2013; Abram et al., 2014; Dätwyler et al., 2018) and has been shown to be
superior in skill to other methods such as PCA (von Storch et al., 2009). Using this method, proxies are normalised to have a 0
mean, unit standard deviation and then weighted according to their correlation to the model SAM over the calibration window,
before being summed into a single time-series. To quantify the skill of the pseudoproxy reconstructions, Pearson correlation
coefficients are calculated between each SAT/precipitation-derived SAM index and the sea level pressure-derived SAM index.

| Region | Window size | Number of sites | |
| --- | --- | --- | --- |
| | | SAT | Precip |
| | 31yrs | 842 - 1740 | 549 - 935 |
| S. Hemisphere | 61yrs | 640 - 1568 | 429 - 709 |
| | 91yrs | 838 - 1535 | 326 - 660 |
| | 31yrs | 557 - 1346 | 264 - 563 |
| Antarctica | 61yrs | 454 - 1253 | 191 - 403 |
| | 91yrs | 705 - 1254 | 211 - 396 |
| | 31yrs | 48 - 152 | 60 - 166 |
| Aus/NZ | 61yrs | 41 - 130 | 46 - 165 |
| | 91yrs | 31 - 132 | 33 - 158 |
| | 31yrs | 54 - 244 | 62 - 195 |
| S. America | 61yrs | 44 - 227 | 46 - 156 |
| | 91yrs | 30 - 207 | 39 - 154 |

**Table 1.** The range of number of sites available for selection into a SAT or precipitation proxy network for each region and calibration
window size. The range is calculated across the 10 different calibration windows used when creating a network, as discussed in Section 2.3

Finally, running correlations of SAT/precip and the SAM are correlated with the model nino3.4 index to investigate the
role the El Nino Southern Oscillation (ENSO) may play in modulating the pseudoproxy-SAM teleconnection. Each gridcell
is correlated with SAM over a 31, 61 and 91 year running window while the n3.4 index is band-pass-filtered using the same
window size to remove high-frequency variability. The two time-series are then correlated over their common interval (500yrs-
window size/2) with significance calculated using a reduced degrees of freedom method (Davis, 1976).





# 3 Results

The importance of a long calibration window is illustrated in Figure 2. The x-axis in all panels shows the 'true' correlation (i.e, calculated over the full 500 years) between the model SAM and either SAT or precipitation, while the y-axis shows how this

'true' correlation may change when that same variable is correlated with SAM with a 31, 61 or 91 year window, rather than the full 500 years. For example, a true correlation of -0.3 between precipitation and the SAM may become anything ranging from -0.65 to 0.1 when evaluated over just a 31 year segment (Figure 2d). However, as the window size increases, it is increasingly likely that the calculated correlation is representative of the true correlation. This is clear from Figure 2a and c (or Figure 2d and f), where the use of a 91 year correlation window means that the range of possible correlations is closer to the true

correlation than for the 31 year window. Also noteworthy is the considerable decrease in the maximum available number of proxies eligible for inclusion in reconstructions when calibrating with a 61 year window, rather than a 31 year window (Table 1). A smaller decrease in the proxy pool is seen when lengthening the window from 61 to 91 years.

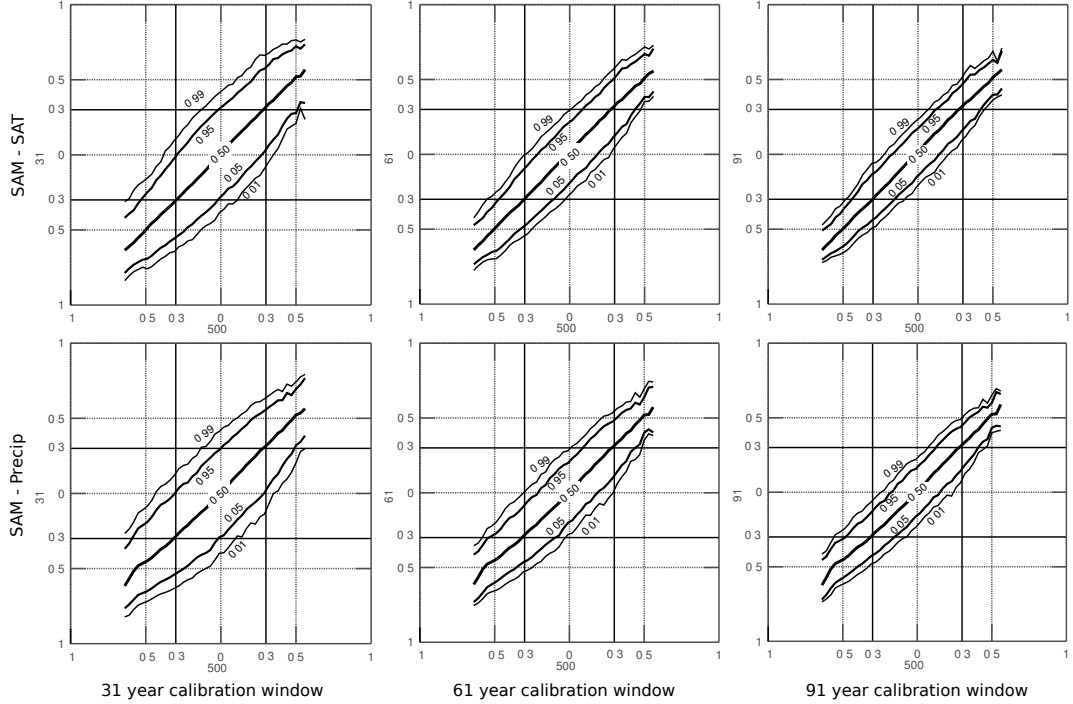

**Figure 2.** Correlation coefficients between the model SAM index and both SAT (a, b and c) and precipitation fields (d, e and f). Panels show the probability distribution of a gridpoint having a certain probability in a 31 (a and d), 61 (b and e) and 91 (c and f) year calibration window, given that same point's correlation over the full 500 years. This illustrates how a longer calibration window will ensure that the correlation of a point to the SAM within that window will be closer to the 'true' correlation, calculated over the full 500 years.



### 3.1 Reconstruction skill

Reconstructions of the SAM often rely heavily on proxies from a very limited geographic location. What follows are several

reconstructions, each one utilising proxies from one or more regions which are commonly used to reconstruct the SAM. To quantify skill, we define a skillful reconstruction as one that is able to reproduce at least 50% of the model SAM variability (i.e., $r^2 \geq 0.5$ or $r \geq \sim 0.71$). In the first scenario, pseudoproxies are sourced from the entire Southern Hemisphere (SH), including Antarctica and this is shown in Figure 3a-f. The reconstruction skill is displayed as a correlation (y-axis) between the pseudoproxy generated SAM index and the 'real' SAM index calculated from sea level pressure (SLP) fields in the model.

This is plotted against the number of proxies used to generate the reconstruction (x-axis) and the range in the 5th, 50th and 95th percentiles represents the use of 10 different calibration windows and the effect this has on the reconstruction.

Results suggest that small proxy networks (2-10 records) rarely provide skillful reconstructions of the SAM, even when calibration is a relatively large 91 years (Figures 4 and 5, panels a,e,i - red line), though a greater proportion of precipitation-derived reconstructions are considered skilful across all window sizes. The range in reconstruction skill is smaller for precipitation than

for SAT, particularly when longer calibration windows are used, suggesting larger multi-decadal variability in the SAM-SAT teleconnection over time. Maximising the number of records in the proxy network leads to a larger proportion of skilful reconstructions, although for the shortest window of 31 years, the reconstruction skill in the $95^{th}$ percentile is never greater than $r = 0.76$ for precipitation and $r = 0.77$ for SAT ($r^2 = 0.58$ and 0.59 respectively), suggesting that at most around 60% of the model SAM variability can be reproduced. Minimum values (lowest $r$ value in the $5^{th}$ percentile for 70 proxies) are $r = 0.62$

for SAT and $r = 0.59$ precipitation, reconstructing 38% and 35% of SAM variability, respectively (Figure 3a and d).

The range in reconstruction skill in Figure 3 is a result of the use of 10 different calibration windows over the 500 year period and indicates that, even when the network size is maximised and a long window is selected, simply calibrating during a different period can change the skill of the resulting reconstruction. It is noteworthy that increasing the calibration window length does not necessarily increase the maximum possible skill of the resulting reconstruction, but rather leads to a reconstruction

converging towards a so-called 'perfect' reconstruction. This 'perfect' reconstruction has a 500 year calibration and shows the actual ability of these proxies to reconstruct the SAM. This convergence is visible for SAT reconstructions (Figure 4a,e, and i). In other words, a longer calibration window will more realistically represent a proxy's relationship with the SAM but as a result, may decrease the 'skill' of the reconstructed SAM.

In an attempt to represent tree-ring-only reconstructions such as that of Villalba et al. (2012), reconstructions are included

for proxies sourced from the entire Southern Hemisphere other than Antarctica. The Antarctic-free SAT reconstructions are less skilful for the 31 and 61yr windows with a larger range. Of note is that most of the $95^{th}$ percentile (Figure 3g, h, i - red shading) is below the $r^2 > 0.5$ skilful threshold, as opposed to reconstructions with Antarctic sites. Antarctic-free precipitation reconstructions typically see an increase in maximum skill but a similar increase in the range (Figure 3i-l). The contrasting effects of Antarctica could be due to Antarctic precipitation having a generally weak correlation with the model SAM in

31, 61 and 91 year running windows, while SAT shows strong negative correlation with SAM continent-wide (Figure 1a)

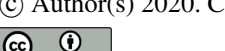



**Figure 3.** Correlation of the model SAM index (y-axis) to the pseudo-proxy reconstructions described in Section 2.3, plotted here by network size (x-axis). This is therefore a measure of how skillfully each proxy network is able to reconstruct SAM. Each panel shows reconstruction skill for the 31, 61 or 91 year running windows. The three shaded areas show the $5^{th}$, $50^{th}$ and $95^{th}$ percentile. Their range represents the range of these percentiles across the 10 different calibration windows for each window/ network size. Panels a-c and d-f show reconstructions for SAT and precipitation respectively, where proxies are sourced from the entire Southern Hemisphere. Panels g-i and j-l show reconstructions generated by proxies sourced from everywhere but Antarctica, but are otherwise equivalent to a-f.

and by removing these points, we lose skill in the SAT-derived reconstruction and increase skill in the precipitation derived reconstruction.

   Data from different regions may also act to increase or decrease the skill of reconstructions. Figures 4 and 5 illustrate the skill of each regional reconstruction in comparison to the Southern Hemisphere (SH) one. In addition, comparisons are made



to a reconstruction with an 'ideal' calibration window of 500 years, showing the actual range in skill that the pseudoproxies can produce. When we utilise records from individual regions the reconstructive skill of the proxy network is significantly reduced. Reconstructions for the Australia-New Zealand region (Figure 4 and 5c,g,k), South America (Figure 4 and 5b,f,j) and Antarctica (Figure 4 and 5d,h,l) all show reduced reconstructive skill when compared with the entire SH network (Figures 4 and 5a,e,i), with Antarctica being the only individual region capable of generating any skillful reconstructions. In general

then, reconstructing the SAM using pseudo-proxies in CM2.1 is most successful when we maximise network size and source sites from as many geographical regions as possible, particularly at longer calibration windows, where the proxy pool becomes too small for a full network in many regions. Southern Africa was excluded from this analysis as too few grid cells met our criteria for reconstruction. The exceptions here are precipitation-based reconstructions, where leaving out Antarctica improves reconstruction skill.

When comparing proxy types, there are significantly more 'skilful' precipitation-derived reconstructions than for SAT, and this is true across all window sizes for the Southern Hemisphere-wide reconstructions (Figures 4 and 5a,e,i). In particular, when using a 61 or 91 year window, 89% and 91% of SH precipitation reconstructions are considered 'skilful', respectively (with a network size of 70)(Figure 5e and i) and an increase in window size increases the proportion of skilful reconstructions (Figure 5a,e,i; red line). In contrast, SAT reconstructions calibrated with 61 and 91 year windows only produce skilful reconstructions

25% and 20% of the time, respectively (Figure 4e and i). Most striking here is that a longer calibration window both decreases the $95^{th}$ percentile skill and the proportion of SAT-derived reconstructions that can be considered skilful. But this is reasonable when we see that, at best, 11% of 'perfectly' calibrated SAT reconstructions are skilful and have a lower maximum skill for the $95^{th}$ percentile (Figure 4e and i). This indicates that shorter calibration windows are sufficiently susceptible to climatic noise or modulation that they are producing reconstructions with spuriously larger reconstruction skill. It is also worth noting that the

reduction of reconstruction skill range visible for the 61 and 91 year windows relative to the 31 year window will necessarily be in part due to the overlapping of the 10 calibration windows over the 500 years of model data. With a longer data set, the lack of such an overlap would almost certainly result in this spread being larger.

   While increasing the number of sites used in each reconstruction does not necessarily improve the maximum $95^{th}$ percentile skill after approximately n = 10, it does narrow the range of possible reconstruction skill (Figures 4 and 5a,e,i; note the yellow

envelope converging on the blue with increasing window size). It should be noted that while Figures 4(a,e,i) give the impression of our reconstructions outperforming the 'perfect' reconstructions, they have virtually the same maximum skill at each network size. This apparent incongruence occurs due to the probability distribution of reconstruction skill for our 'perfect' proxies being far narrower than for the reconstructions with varying window length, resulting in the $95^{th}$ percentile having a generally lower value for each network size.

## 3.2   Mapping non-stationarity

In this section we examine whether certain regions are more or less non-stationary to SAM, which would contribute to these regions being better or worse than others at reconstructing SAM. Figure 6 shows the number of non-stationary years at each grid point for SAT (a) and precipitation (b) as defined in Section 2.2. Grid points with running correlations that fall outside the





**Figure 4.** Differing reconstruction skill achieved when using SAT-derived proxies sourced from the entire Southern Hemisphere (a, e, i), South America (b, f, j), Australia and New Zealand (c, g, k) and Antarctica (d, h, l) only. The correlation between a SAT-derived reconstruction and the SAM is on the y-axis, while the number of sites used (n = 2:70) in a reconstruction is on the x-axis. Shaded regions represent the range between the minimum of the $5^{th}$ percentile and the maximum of the $95^{th}$ percentile for each network size, across 10,000 reconstructions (described in Section 2.3). Each set of regional reconstructions is shaded in yellow and the end of this yellow region indicates the number of samples available when it is below 70. Each panel also includes the range in skill for reconstructions with sites sourced from the entire Southern Hemisphere (SH) and calibrated with an 'ideal' 500 year window (blue shading). The blue line indicates the percentage of ideal SH reconstructions that meet or exceed our skill threshold of being able to explain 50% or more of the variability in the SAM. The red line indicates the same thing, but for each regional reconstruction. The dashed black line indicates the $r$ value required to meet our skill threshold.

95% confidence interval of stochastic variability more than 10% of the time are highlighted with solid contours; we define these
regions as non-stationary. Depending on the length of the calibration window, different patterns of non-stationarity appear, par-





**Figure 5.** Differing reconstruction skill achieved when using precipitation-derived proxies sourced from the entire Southern Hemisphere (a, e, i), South America (b, f, j), Australia and New Zealand (c, g, k) and Antarctica (d, h, l)) only. The correlation between a precipitation-derived reconstruction and the SAM is on the y-axis, while the number of sites used (n = 2:70) in a reconstruction is on the x-axis. Shaded regions represent the range between the minimum of the $5^{th}$ percentile and the maximum of the $95^{th}$ percentile for each network size, across 10,000 reconstructions (described in Section 2.3). Each set of regional reconstructions is shaded in yellow and the end of this yellow region indicates the number of samples available when it is below 70. Each panel also includes the range in skill for reconstructions with sites sourced from the entire Southern Hemisphere (SH) and calibrated with an 'ideal' 500 year window (blue shading). The blue line indicates the percentage of ideal SH reconstructions that meet or exceed our skill threshold of being able to explain 50% or more of the variability in the SAM. The red line indicates the same thing, but for each regional reconstruction. The dashed black line indicates the $r$ value required to meet our skill threshold.



ticularly for SAT. Aside from three small regions in south-east Australia, central South America and the Queen Elizabeth Range in Antarctica, there are almost no land sites that can be considered non-stationary when using a 31 year running correlation for SAT. It is also noteworthy that these non-stationary regions (as calculated using the 31 year running correlation) appear to fall, on average, in regions where correlations are weaker over the full 500 years (Figure 1a and c). The same is also broadly true

for precipitation, where large regions of non-stationary points do occur but fall in regions of weaker or 0 correlation with the SAM, particularly in East Antarctica. It is worth noting, however, that despite not meeting the requirement of being classified as non-stationary, large regions of the Southern Hemisphere land surface show modulation of the SAM-proxy teleconnection.

To better illustrate the impact of non-stationary proxies on reconstructions, Figure 7 shows the proportion of reconstructions that contain a certain percentage of non-stationary sites. Perhaps unsurprisingly, smaller network sizes have a greater chance

of having a high proportion of non-stationary sites, regardless of calibration length. In spite of this, a network containing only 2 sites calibrated with a 31 year window still only has a 0.2% chance that both sites will be non-stationary (Figure 7a) and the odds of a network containing a large proportion of non-stationary sites diminishes greatly once it contains 30 or more sites (Figure 7a and b). Overall, when using the 31 year running window on the precipitation data over land only, 136 points - or 6% of land points - are defined as non-stationary. For SAT this equates to 91 points, or 4% of land points. For the sake

of completeness, we show similar results for the 61 and 91 year calibration window (Figure 7b,c and e,f respectively) which demonstrate that a longer calibration window results in more non-stationary proxies being selected for reconstruction. This is consistent with the number of non-stationary points increasing when using a 91 year running correlation: 205 points - 9% of land points for precipitation and 176 points or 8% of land points for SAT. In summary, while increasing the calibration window results in a larger proportion of non-stationary sites in a reconstruction, a sufficiently large proxy network minimises

the probability that these non-stationary sites will represent a significant proportion of the network.

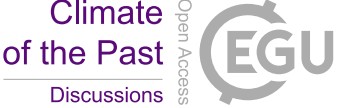



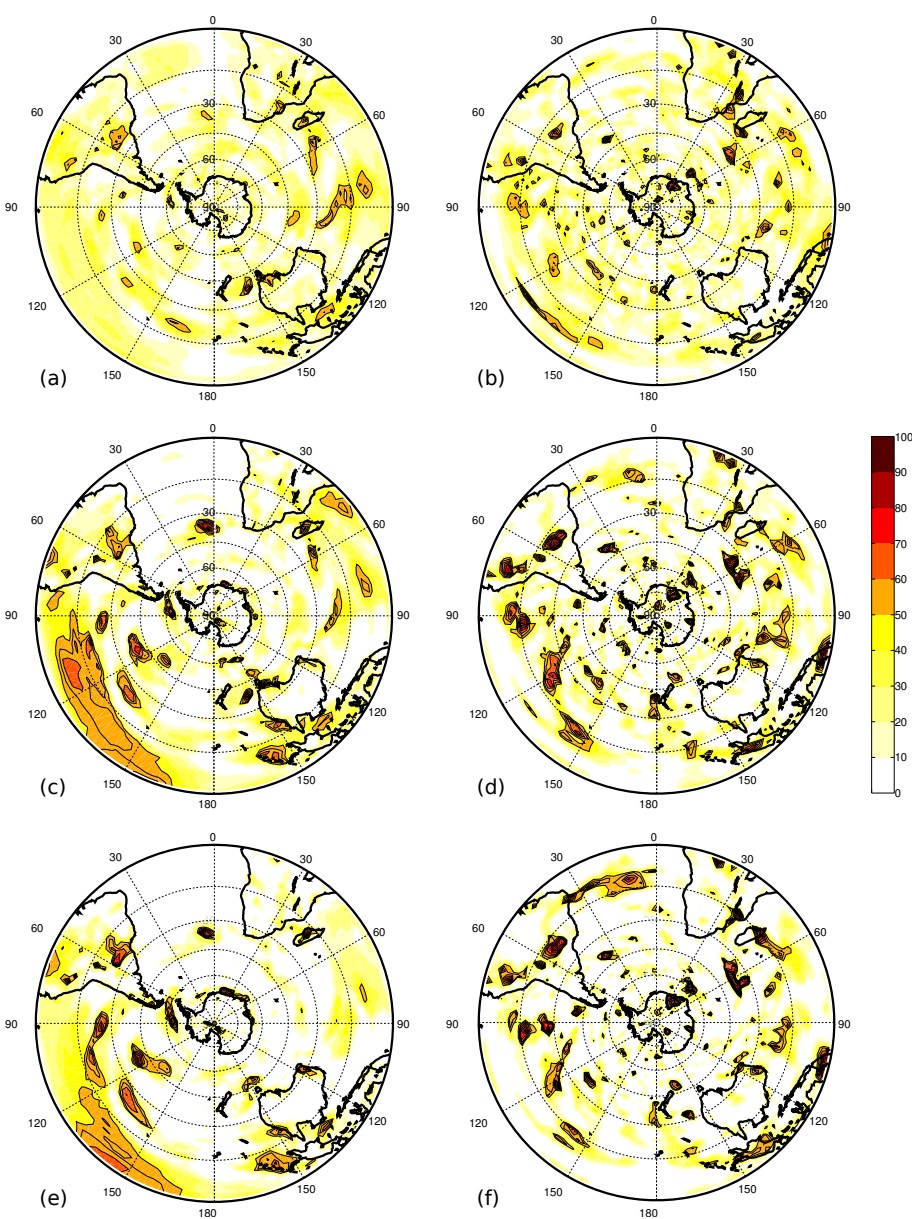

**Figure 6.** Number of years at each gridpoint where the 31 year (a,b), 61 year (c,d) and 91 year (e,f) running correlation between SAT (a,c,e) or precipitation (b,d,f) and the model SAM falls outside the 95% 'stationarity' confidence interval (Section 2.2). As per our definition of non-stationarity, regions which fall outside this interval 10% of the time or more ($\geq$ 50 yrs) are highlighted with solid black contours and are considered to be non-stationary.

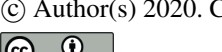



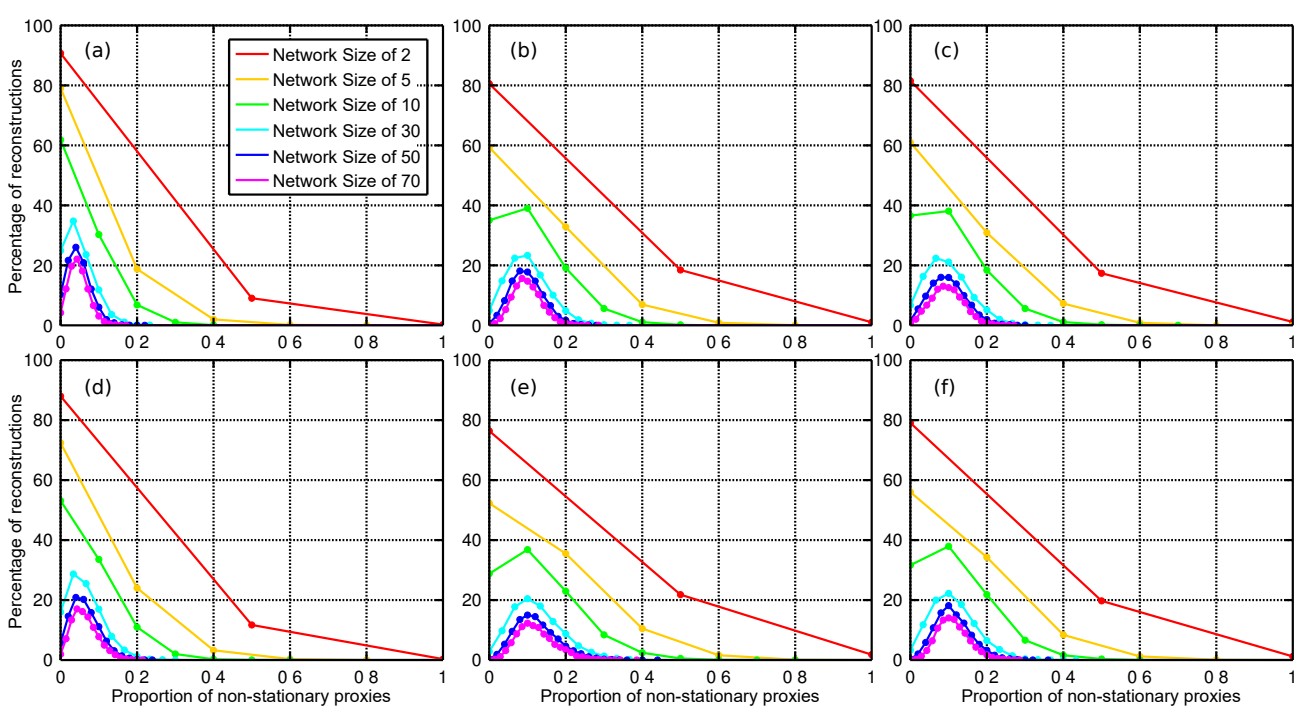

**Figure 7.** The chance of creating a SAM reconstruction (y-axis) with a certain proportion of non-stationary proxies (x-axis) as calculated from 10,000 reconstructions for each network size. Panels show probabilities for 31 (a and d), 61 (b and e) and 91 (c and f) year calibration windows. Panels a-c and d-f show reconstructions based on SAT and precipitation data respectively.





### 3.3 Modulation of the SAM-proxy teleconnection

While few terrestrial cells qualify as non-stationary based on the definition in Section 2.2, there is still considerable variance in the teleconnection strength between SAM and SAT/precipitation over the 500 years of the simulation (Figure 8). While this could be due to climatic noise, it is not unreasonable that other modes of climatic variability - in particular, ENSO - may be
modulating this teleconnection (Silvestri and Vera, 2009; Fogt et al., 2011; Dätwyler et al., 2020). The regions from which we source our proxies, such as Australia/ New Zealand and South America are strongly impacted by ENSO oscillations, visible in both temperature and precipitation fields (Davey et al., 2014). The following section will examine which regions show the most variance in proxy-SAM teleconnection and whether or not these regions appear to be influenced by the model ENSO.

Teleconnection strength for SAT proxies with SAM shows considerable variance in Antarctica, northern and southern South
America and for the 31 and 61 year windows, parts of Australia and New Zealand (Figure 8a-c). Distinct regions of higher variance in Antarctica are typically in regions of high SAM-SAT correlation over the full 500 years (Figure 8a-c; dashed contours), with this variance decreasing as we move to a 91 year calibration window. Variance is low, or no data is present, for regions with a small 500yr $r$ value, though this is in part due to our correlation criteria when selecting proxies.

Significant correlations between the running correlations of SAM-SAT and the filtered nino3.4 (n34) index can be seen over
much of Western Antarctica, south-eastern Australia and parts of South America (all windows, Figure 9,a-c) while the 31 year window also shows significant correlations in East Antarctica. ENSO's modulating influence can be seen to vary, depending on both the strength of the underlying SAM-proxy correlation as well as the calibration window length (Figure 11). For the 31yr window, the regression coefficient between ENSO and the SAM-SAT running correlation is relatively low, clustering predominately between 0.2 and 0.4 (Figure 11a), and generally decreases if a site has a stronger correlation with SAM over
the full 500 year period. A similar relationship is visible for the 61yr window, however the regression coefficients are slightly larger ($\sim 0.25 - 0.5$) and the decrease in ENSO regression coefficient with increase in SAM-SAT $r$ is less apparent. The 91yr window sees this relationship disappear altogether, with relatively large ($\sim 0.6 - 0.8$) regression coefficients independent of SAM-proxy correlation (Figure 11c and f). This is of lesser consequence, however, as most sites show little variance in SAM-SAT teleconnection at this longer window length (Figure 8c; most regions have an $r_{std} < 0.1$). To the extent that ENSO can
modulate the SAM-SAT telconnection, this can be reduced with a longer calibration window as the number of land points (SAM-SAT running correlation) significantly correlated with the filtered n34 index is 30%, 24% and 15% for the 31, 61 and 91yr windows respectively. Failing this, a high SAM-SAT correlation coefficient at shorter window lengths will decrease the likelihood that ENSO will more strongly modulate the teleconnection.

For precipitation, teleconnection strength is less variable and only parts of Australia, Indonesia and the Ross Ice Shelf/ Marie
Byrd Land in Antarctica show large changes (Figure 8d-f). Correlation of this precipitation teleconnection variance with the model nino3.4 index reveals few regions of significant ENSO influence (Figure 10) and little coherent spatial structure for this correlation. The magnitude of the impact of ENSO on the SAM-precip teleconnection is similar to that of SAT proxies (Figure 11d,e,f). The number of grid cells impacted by ENSO is fewer than that for SAT, with the running correlation of SAM-precip being significantly correlated with n34 in 21%, 16% and 8% of land cells for the 31, 61 and 91 year windows.





**Figure 8.** One standard deviation of correlations between SAT (panels a-c) and precipitation (panels d-f) with the model SAM index, over the 10 calibration windows. Panel a) shows values for the 10, 31 year calibration windows for SAT. Panels b) and c) are for the 61 and 91 year windows respectively. Panels d), e) and f) are the same, but for precipitation. Maximum values on colourbars for panels a),b),d),e) and f) indicate the colour of several outliers in the data. Grey regions indicate cells that did not meet the minimum correlation criteria (r >= +-0.3) over 8 or more windows, meaning no standard deviation could be calculated. Dashed contours show the model SAM-SAT (a-c) and model SAM-precipitation (d-f) 500 year correlation fields from Figure 1.



**Figure 9.** Correlation between the (a) 31yr, (b) 61yr and (c) 91yr SAM-SAT running correlation at each gridcell and the model-derived n34 index. The n34 index is filtered with a corresponding 30, 60 and 90 year filter. Hatched regions indicate p < 0.05.

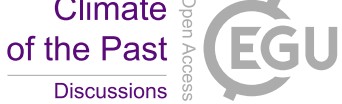

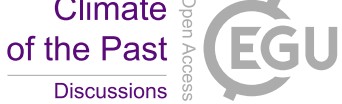

**Figure 10.** Correlation between the (a) 31yr, (b) 61yr and (c) 91yr SAM-precipitation running correlation at each gridcell and the model-derived n34 index. The n34 index is filtered with a corresponding 30, 60 and 90 year filter. Hatched regions indicate p > 0.5.





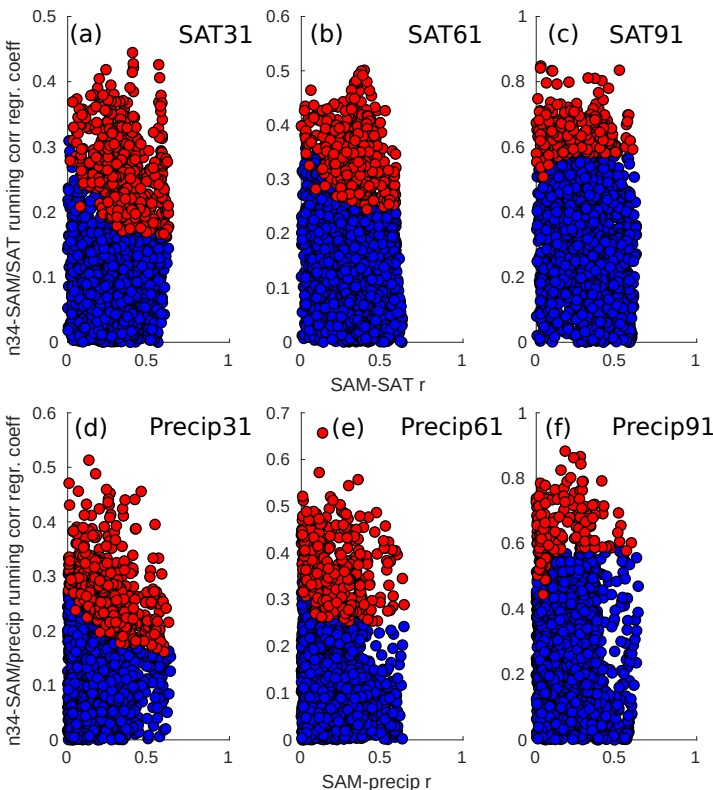

**Figure 11.** Scatter plot of the regression slopes from the significant (red) and non-significant (blue) land points shown in Figures 9 (panels a,b,c) and 10 (d,e,f) plotted against the 500yr correlation coefficient between SAM and SAT (a,b,c) and precipitation (d,e,f). Both SAM-proxy running correlations for each cell and the n34 index are standardised prior to regression calculation. Note the changing vertical scales between panels.

## 4  Discussion and Conclusions

In this study we use the GFDL CM2.1 coupled climate model to examine how regional biases in sourcing of proxy data impact on the reconstructive skill of the SAM, as well as whether or not non-stationarities in these proxy networks significantly impact the reconstruction. Reconstructions derived from model surface air temperature (SAT) and precipitation fields and calibrated over a 31 year window are able to - at best - replicate 56% to 58% of SAM variance respectively, comparing favourably to an 'ideal' proxy calibrated over the full 500 year interval (Figures 4 and 5).

Assessing the skilfulness of our reconstructions, where skilfulness is defined as being able to reproduce >= 50% of SAM variance over the full 500 years, those derived from precipitation performed best, with a maximum of 91% of reconstructions being reported as skilful (91yr window, 70 proxies). SAT-derived reconstructions performed poorly by comparison, with only



25% qualifying as skilful (61yr window, 70 proxies). This is in line with Gallant et al. (2013), who show that precipitation
appears to be more skilful not only in the Australia/New Zealand region but globally as well (Figure 3) with higher maximum
reconstructive skill, a higher proportion of skilful reconstructions as well as less spread due to variability of the teleconnection
between precipitation and the SAM.

Both SAT and precipitation-derived reconstructions were most skilful when proxies were selected from a geographically
broad region, while regional reconstructions - with the exception of Antarctica - fail to produce any skilful reconstructions.
This is likely due to each region being affected by localised climatic noise which becomes a systematic source of error in the
reconstruction. With larger datasets from different regions, this noise cancels out and the signal we seek to reconstruct is more
clearly visible.

Increasing the calibration window does not increase the chance of producing a more skilful reconstruction, it does however,
along with maximising the number of proxies, cause the range of reconstruction skill to converge on the skill of our 'perfectly'
calibrated proxy reconstructions (blue envelopes, Figures 4 and 5). Adding more sites to a reconstruction has limited benefit in
terms of the maximum skill it can achieve, with values largely plateauing at a network sizes of $\sim$20. When it comes to minimum
skill, however, this improves for increases in network size all the way up to and including network sizes of 70 proxies (Figure
3). This in turn, acts to increase the proportion of skilful reconstructions for a given window size.

Low frequency variability exists in the teleconnections between our pseudoproxies and SAM that cannot be explained by
climatic noise (stochastic variability). CM2.1 simulates, at maximum, 9% of land points as being non-stationary as defined
by Gallant et al. (2013) (using precipitation as a proxy and a 91 year running window), although the odds of creating a proxy
network with a high proportion of non-stationary sites remains relatively low (Figure 7). However the spread in possible
reconstruction skill suggests that this definition of non-stationarity does not adequately capture the variance in SAM-proxy
teleconnection strength and its impact on reconstructions. For the shorter calibration window of 31 years, the variance of the
correlations can be of the order of our cutoff criteria ($r>=|0.3|$) despite not qualifying as non-stationary. While assessing the
stationarity of a proxy-SAM correlation is more difficult in the real world, we suggest that multiple methods be employed
where possible.

It is not unreasonable to suspect that ENSO may be contributing towards the proxy-SAM teleconnection variance. Dätwyler
et al. (2020) identify a highly variable, but centennial-average $r$ of -0.3 between austral summer ENSO and SAM reconstruc-
tions over the last millennium. Their pseudoproxy experiments using a CESM1 ensemble show significant changes in SAT
during periods of large negative SAM-ENSO correlation (their Figure 4, bottom left panel). The pattern is similar to our results
(Figure 9a), with regions of significant correlation over much of Antarctica and three regions in the Southern ocean centered
on roughly 60°E, 150°E and 60°W. To the extent that ENSO impacts our reconstructions, this can be reduced by 1) using
a 91yr calibration window as opposed to a 31 year window, halving the number of points whose SAM-proxy teleconnection
significantly correlate with ENSO (15% vs 30%); 2) Avoid sourcing proxies with significant correlation to ENSO and 3) se-
lecting proxies with strong correlation to SAM, minimising ENSO's impact. Importantly, these results are almost certainly
different for seasonal data. The SAM-ENSO relationship is highly seasonal (Fogt and Marshall (2020) and references therein),
particularly in Austral spring and summer, seasons which typically provide the most SAM paleoarchives such as tree rings.





On the other hand, Yun and Timmermann (2018) demonstrate that a simple linear stationary stochastic perturbation process
can explain low frequency modulation between ENSO and Indian summer monsoon rainfall, requiring no external modulation
via another climatic mode. If a similar principle applies to the modulation of SAM and temperature/ precipitation fields, these
changes in teleconnection strength may simply be the result of climatic noise.

As we use only one integration from a single model, it is worth discussing the performance of CM2.1. Its representation of
the SAM is good with respect to similar models (Karpechko et al., 2009; Marshall and Bracegirdle, 2015), though it does have
some biases which may impact the results presented here. For instance, when compared to observations and reanalysis, there is
a small equatorward bias in the Southern Hemisphere westerlies in CM2.1, however, the spatial structure and amplitude of SLP
anomalies associated with these winds, and therefore the SAM, are well simulated (Delworth et al., 2006). Assessing these
results in individual seasons was beyond the scope of this study, in part due to the CMIP5 generation of models (including
CM2.1) being less skillful at representing seasonal variability in the SAM-SAT relationship than over the annual-mean (Mar-
shall and Bracegirdle, 2015), in addition to the hope that reconstructing the SAM on an annual-mean time-scale would smooth
out high-frequency noise and enhance the signal to noise ratio of our reconstructions.

Overall, CM2.1 appears to replicate the spatial structure of correlation between our SAM index and temperature and pre-
cipitation seen in observations and reanalysis on annual-mean time-scale. The SAT signal (Figure 1a) closely resembles the
pattern observed in the ERA-Interim reanalysis (Figure 1b), including the strong anti-correlation on the Antarctic coast from
60°E-120°E, the positive correlation in the Drake Passage sector and hints of the negative correlation that the model produces
over the Australian continent, further corroborated by Gillet et al. (2006). Similarly, the precipitation correlation map shows
good spatial agreement with the ERA-Interim. CM2.1 reproduces the dipole-like pattern of high (Figure 1c, 60°S-70°S) and
low (40°S-50°S) correlations observed in ERA-Interim as well as the region of negative correlation over the Ross Sea region
(Figure 1d). These comparisons are encouraging, particularly considering the multi-decadal changes in the teleconnections that
have been observed from in-situ temperature and precipitation measurements (see Silvestri and Vera (2009), Figure 1; Gillet
et al. (2006), Figure 1).

A caveat of this study is our use of annual mean data, rather than seasonal fields. This is a distinction from previous real-
world reconstructions utilising tree ring records (Zhang et al., 2010; Villalba et al., 2012; Abram et al., 2014; Dätwyler et al.,
2018) which are not only more sensitive to SAT or precipitation of a particular season, but also combine these with other
proxies such as ice cores (Zhang et al., 2010; Abram et al., 2014; Dätwyler et al., 2018), corals (Zhang et al., 2010) and lake
sediments (Abram et al., 2014; Dätwyler et al., 2018) each of which may be more or less seasonally sensitive. In addition,
many proxies such as tree rings (Cullen and Grierson, 2009; Villalba et al., 2012) have been shown to have a lag relationship
with SAM from the previous year, which is also not accounted for in this study. Dätwyler et al. (2020)'s 'perfect' pseudoproxy
experiments for an austral summer SAM show similar reconstruction skill to our results (an average 31yr running correlation
of ~0.7-0.8 for their ensemble mean) and while the methods of this study are not analogous to theirs, it supports the conclusion
that proxy-derived reconstructions of the SAM in a model framework can, at best, reproduce 50-60% of SAM variance on an
annual time-scale.

Most importantly, our results confirm that calibrations of paleo-data to instrumental records over brief time periods can result in misleading teleconnection strengths. The large range in reconstruction skill due to the use of multiple calibration

windows suggests real life proxy data may provide a misleading representation of reconstructed SAM variability due to this non-stationary behaviour, particularly when the reconstruction network constitutes fewer than 20 proxies. For a SAT-derived, Southern Hemisphere wide reconstruction, a longer calibration window minimises this uncertainty, but doesn't necessarily result in a more skillful reconstruction. Rather, the reconstructions converge on the 'true' skill that these proxies provide. The use of teleconnection stability screening as utilised in Dätwyler et al. (2018) is an important step in the right direction,

and should be utilised alongside correlation 'skill' scores to assess the reliability of a reconstruction. The lack of long term observational data makes it difficult to circumvent this problem, however climate models which have demonstrated realistic dynamical mechanisms may aid us in calculating the uncertainty of these calibrations in the future.

Model data was downloaded from ftp://nomads.gfdl.noaa.gov/gfdl_cm2_1/CM2.1U_Control-1860_D4/pp/, last accessed

30/08/20. The Marshall SAM index was downloaded from http://www.nerc-bas.ac.uk/public/icd/gjma/newsam.1957.2007. seas.txt, last accessed 9/10/20. ERA-Interim data was downloaded from https://www.ecmwf.int/en/forecasts/datasets/reanalysis-datasets/ era-interim, last accessed 9/10/20. Code for analysis and plotting of figures can be found in the following Github repository: https://github.com/whuiskamp/SAM_pseudoproxy

Analysis and plotting was done using Matlab v2017b, Pyferret v7.4 (PyFerret is a product of NOAA's Pacific Marine Envi-

ronmental Laboratory. http://ferret.pmel.noaa.gov/Ferret/) and python3.

*Author contributions.* W.H. and S.M jointly conceived the study. W.H. performed the analysis and created all figures. Data interpretation was done by both authors. The manuscript was written by W.H. with input from S.M.

*Competing interests.* The authors declare that they have no competing interests.

*Acknowledgements.* The author's would like to thank Ryan Batehup for the use of his code (Batehup et al., 2015) which was substantially

utilised in this study. W.H. was supported by the Australian Research Council (ARC) via grants FT100100443, DP130104156 and the Potsdam Institute for Climate Impact Research (PIK), member of the Leibniz Association. S.M. was supported by the Australian Research Council (ARC) via grant number FT160100162.




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
