# Peer review of "Quantifying paleo-reconstruction skill of the Southern Annular Mode in a model framework"

_Climate of the Past, 2020_

## Referee Comment (RC1) · Anonymous Referee #1 · 18 Nov 2020

article [utf8]inputenc hyperref

[Figure]

**Review of Huiskamp and McGregor**

November 2020

**1 Overview**

This study is a pseudo-proxy experiment, based on the GFDL CM2.1 model, which explores the experimental sensitivity of hemispheric-wide reconstructions of the SAM index. I have personally tried to perform reconstructions of the historical SAM index in a "real-world" experiment and I was not convinced that it's currently feasible because of several limitations (trend in SAM, too few *in-situ* observations, too few continental proxies in SH,...). I thus think that the present study is very relevant within this context, since reconstructing SAM is very challenging. Indeed, I am convinced that the use of GCM(s) is essential for addressing the limitations in real-world reconstructions of SAM. I appreciate the way the authors have tried assessing different sources of uncertainty for the reconstruction such as non-stationarity in SAM-proxies relationships, the length of the calibration window and the size of the (pseudo-)proxy network. I also appreciate that the manuscript is well-written and generally easy to follow with a clear story-line.

However, there are some aspects in this study, that have poor statistical meaning. I would like that the authors try to address the different comments/suggestions below. Although I propose an "Accepted manuscript with minor revisions", these revisions might need running again some codes and slightly modifying some figures. However I am pretty sure that the suggestions I make won't bring large modifications in authors conclusions/discussions.

**2 Comments/Suggestions**

**2.0. Abstract**

- The authors should mention the complete acronyme of the model, *i.e.* GFDL CM2.1, at leat in the abstract. More generally, the authors go back and forth between "CM2.1" and "GFDL CM2.1" in the main text. Two options:
    1. They should mention it once in the main text followed by brackets mentioning the short appellation : "[...] GFDL CM2.1 (CM2.1 hereafter) [...].
    2. Otherwise they should call it with the long name in the whole text and figures. Since the model is called "CM2.1" in figures, option 1 seems the less constraining.

**2.1. Introduction**

- Figure 1: Maps are only partially informative since no confidence levels for correlations are shown. Instead, the authors use contours for $|r| > 0.3$, and I guess the other levels of contours are drawn every 0.1 steps (should have been mentioned in caption if that's the case). I understand these contours are useful when using continuous colorscales, I personally don't like it but that's not an issue here. However, the authors should at least add on maps which correlations are significant, and which are not (at the 90% confidence level, for instance). Indeed, for model simulations and instrumental data used here, time frames over which correlations are calculated have very different lengths (500 and 36 respectively), such that 0.3 correlation (for example) yields very different levels of significance for the two cases. For instance, the Student test from McCarthy et al. (2015) with corrected degrees of freedom can be used (see their Methods section): https://www.nature.com/articles/nature14491. I understand the authors are not showing significance levels to facilitate the comparison between observed and simulated patterns of SAM, which are indeed well matching, but the significance levels might hide this. The authors should at least address 1), and i suggest that using 2) might give a better figure, but i let the latter up to the authors:
1. The authors should keep colors and just indicate (for example) with little crosses the grid points for which correlations are not significant and explain that the 36 years time frame is too short to capture statistical significance, so we still see that GFDL CM2.1 seems adequate for testing SAM reconstructions. (See attached Fig. 1: same as Fig. 1b from the manuscript but with little black dots indicating correlations unilaterally not significant at the 90% confidence level, using Student test from McCarthy et al. 2015, see link in above paragraph).

2. The historical significance levels can probably be "boosted" using longer temperature and precipitation products, that will enable to use 21 more common years with SAM observations (1958 to 1978) and give more robust estimations of the SAM precipitation/temperature pattern. There is, for instance, the CRU TS dataset (Harris et al. 2020, https://www.nature.com/articles/s41597-020-0453-3).

- L.32: Typo. Space between "cycle" and ".".

**2.2. Methods**

- L89-91: This is one of the major limitations of the study I think. The authors are working on a control run while using a correlation-based weighting technique (weighted CPS). In the real world, the observed SAM has a trend, which is likely to be due to human activity as mentioned by the authors in the introduction. Thus, since most of the temperature signals also has trends in the real world, one can expect calculating correlations (and then weights here) overestimated for the proxies that has the closer trend to the SAM one. In other words, If this study aims at being a support for real-world SAM reconstructions, the authors should mention as a caveat the fact that contrary to the pseudo proxy experiment, the correlations calculated in a real-world experiment over calibration periods might be artificially overestimated/underestimated because of the specific SAM trend.

- L95: "CM2.1 was selected due to its relatively good representation of the SAM.". Since this is the major reason for which the authors use this model, we need an objective statistical test that shows that spatial correlations of SAM with temperature and precipitation simulated from CM2.1 are significantly similar to those from the observations at a given significance level. This is far more convincing than just "CM2.1 has a relatively good representation" with a subjective use of "relatively". I guess a Monte-Carlo approach using distance between pairs of random correlation maps can do the job. I have no doubts that the statistical test won't affect the model choice of the authors.

- L144: I guess the authors mean "$a_0$ and $a_1$ are regression coefficients [...]", instead of "$a_0 + a_1$ are regression coefficients [...]"

- L167: I finally understand the relevance of contours from Fig. 1. As for the comment in the introduction, the choose of $|r| > 0.3$ is here purely arbitrary. Would we obtain the same results with other arbitrary choices like 0.25 or 0.4? I think a way for the authors to gain in transparency in their methodological set-up is to make a selection of proxies where the correlation significance at a given level is used for selecting them, instead of using the absolute values of correlations, that have different meanings for 31, 61 and 91 window length. I know the authors have been honest and claimed that this choice is arbitrary, but I am wondering if a (pesudo)proxy selection based on a statistical test is also geographically too restrictive. If not, I think the authors should consider this way for selecting (pesudo)proxies.

- L188: This statement is way too strong. Probably that in VonStorch's 2009 study, weighted CPS is better in skill than the PCR method. However that might not be true for other datasets (maybe for the author's dataset who knows). There is indeed a very famous mathematical theorem, named the "No free lunch" theorem, that stipulates that given two statistical methods A and B, we can always find datasets for which A is better than B, and others for which B is better than A (Wolpert 1996, https://www.mitpressjournals.org/doi/abs/10.1162/neco.1996.8.7.1341). With this in mind, the authors should remove this statement and just say that the reconstruction sensitivity to the method is beyond the scope of the study.

- L190-191: Are the authors calculating the skill of the correlation using the 500 years of the control simulation? If yes, this is problematic and it should be fixed, because this means that skill scores for longer calibration time frames will be artificially increased, only by the fact that they are including more data over which the statistical model is built. If not, this should be clarified in the main text because this point is not clear to me.
**2.3. Results**

- L198: Please address my last statement from the Methods section because this might affect the first statement of this section.

- L211-212: Seems a bit arbitrary too to say that a reconstruction is skillful when $r^2 > 0.5$, although this is a relatively constraining criterion. I do not require changes here, but the authors should keep in mind that using more relevant evaluation metrics such as the coefficient of efficiency or the reduction of error (discussed for instance in Macias-Fauria et al. 2012, https://www.glaciology.net/pdf/Macias-Fauria-Dendrochronologia12-$_{persistence.pdf}$) can clearly help for addressing this issue (and probably the last one too).

- L230: Is there any reference for this so-called "perfect" reconstruction approach for calculating skills? I feel like this is an "handmade" skill metric, but please let me know if I am wrong. If I am wrong, the authors should add a reference then. Naively, i would say that the "perfect" reconstruction has 1 correlation with the pressure-based model SAM. Here, the authors' definition of perfect is purely relative to the time period covered by the control simulation. If we add 200 more years of simulation, is the 700-years based pseudoproxy reconstruction "more perfect" than the 500-years one? Considering the last two comments about the metric used and correlation calculated over the 500 years (including calibration), I think that I see why the authors use this "perfect reconstruction" skill metric. Here again, I think using more relevant metrics than correlation could avoid overfitting, handmade metrics, and some arbitrary choices.

- L284: It's good to know that non-stationary relationships tend to fall in regions for which correlations are weak. It would have been even more interesting to see that these correlations are not significant (cf. my comments on Fig 1).

- Figure 10: The authors say hatched points are for $p > 0.5$. I guess they mean $p < 0.05$. Otherwise it has no sense.

**2.4. Discussions and conclusions**

- First two paragraphs: The authors claim that reconstructions derived from temperatures are more skillful. But with real world data, we often use both precipitation and temperature proxies. Is there any reason for not considering the case of a mixing of both types of data? Should at least been mentioned in the discussion.

- L343-344: Why comparing the proportion of skillful reconstructions of SAT and precip for 61-years and 91-years calibration window lengths, respectively?

- L374: 1) Or maybe that 31 years windows are simply too short to robustly estimate the significance of the correlation with ENSO.

- L374: Suggestions 1, 2 and 3 for avoiding an ENSO bias in the reconstruction should be discussed a bit more. Do they mean that we need to wait 91 years of direct SAM observations (at least year 2049 then) that overlaps with at least 70 SH proxies (so, far more than 2049) that all are not significantly correlated with ENSO but all significantly correlated with SAM? That's personally what I concluded from these suggestions.

---

## Referee Comment (RC2) · Christoph Dätwyler (Referee) · 24 Nov 2020

**Review of the discussion paper «Quantifying paleo-reconstruction skill of the Southern Annular Mode in a model framework» submitted by W. Huiskamp and S. McGregor to the journal Climate of the Past**

Christoph Dätwyler, 24 November 2020

**Summary.** In the submitted study, the authors use pseudo-proxy records generated from the GFDL CM2.1 climate model to assess the effect that the number of contributing records as well as their geographic distribution has on the skill of SAM reconstructions. For the pseudo-proxy-generation the surface air temperature and precipitation fields in the model are used. Furthermore, they analyse the (non-)stationarity of their pseudo-proxy records and aim at identifying the influence of ENSO on the relationship between the pseudo-proxies and the SAM index.

**General comment.** In my opinion, the manuscript has the potential to become a helpful contribution toward our understanding on, e.g., how non-stationarities in proxy records can affect palaeoclimate reconstructions of SAM or how sensitive SAM reconstruction are to the influence of ENSO and also other related questions. However, at the current state I have a hard time to see the scientific value the proposed study offers, because the study heavily suffers from several major issues that all need to be addressed before a publication in Climate of the Past should be considered. I try to give hints below how these issues could be addressed.

I am aware of the fact the for any manuscript every reader most likely has his/her own wish-list of things that could be done in addition to what is already presented. So in the major points below I restrict myself to what I consider a conditio sine qua non and do not ask for things that I think would be "nice to have", but are not of crucial importance.

I very much encourage the authors to take the effort to address the raised concerns and revise their manuscript, since the general topics the study touches are of major scientific importance and with the submitted manuscript already a first step has been taken toward a relevant contribution to our understanding of questions related to these topics.

**Recommendation.** I recommend the manuscript to be re-evaluated after major revisions.

**Major points**

**Robustness.** I have major concerns regarding the robustness of the presented results. I do have the impression that the results depend too much on several choices the authors made. Generally speaking, the results must/should not depend too heavily on such choices to be of any significant value. Or if there is a strong dependence on a choice, this choice must be very well justified. Creating a supplementary material file to complement the manuscript would allow the authors to incorporate the results of robustness tests. Before the manuscript should be considered for publication, checking the robustness of the following points is irremissible.

1.1  How strongly are the results model-dependent? It is well-known that climate models struggle to capture high-latitude dynamics. The model they use is already about 15 years old and I wonder whether there have not been any advances since then. Using a model that is as good as possible is of vital importance for this study if any of the conclusions drawn should have a meaning in real world scenarios.
Moreover, the authors also (partly) justify their choice of CM2.1 with Figure 1 and state that there is a spatially good agreement between the correlations (of SAM with SAT and precipitation) in the model and the reanalysis data (L95-97). I don't agree with this statement because if we, e.g., look at SAT and exclude Antarctica then I'd estimate (visually) that about half of the grid points that are over land (i.e. South America, Australasia and Africa) do not even share the same sign. Adding data from one or more additional model, would strongly improve the quality of the paper, as it would allow to assess all conclusions with regards to the choice of model data.

1.2 How much different would the results look if for the pseudo-proxy selection, a different absolute value for the correlation with the model-based SAM index would be chosen? I don't even think fixing an absolute value does make sense at all because different window width for calibration are used. It is much easier to get a high correlation when correlating only over 31 years as compared to 91 years. Rather, the choice whether a record is goes into the reconstruction should be based on whether the correlation with the target is significant or not.

1.3 How much does the choice of $r^2 \geq 0.5$ / $r \geq 0.71$ when defining a "skilful reconstruction" affect the outcome? This choice is very cumbersome to me and I cannot see any justification for it. By just looking at the figures, I suspect choosing e.g. $r \geq 0.6$ or $r \geq 0.8$ would completely invalidate the conclusions that stand in connection with this measure of skill. Given that the authors screen their proxy location using correlations, it is to be expected that a reconstruction rated with a correlation-based skill score will perform rather well. Instead, a skill measure that is different from the condition used for screening would allow a more robust assessment. Furthermore, since this definition of skilful is purely looking at the correlation of the reconstruction with the model-derived SAM index, the reconstruction could theoretically be completely off and still have perfect skill, or the reconstruction could have lost almost all its variance and be nearly completely flat while still having an extremely high correlation.
There are many possibilities that could potentially help here. I am think of accuracy measures such as e.g. RMSE, RE and CE. I hope this suggestion helps the authors to find a justifiable way of measuring skill.

1.4 How strongly do the results depend on the 10%-choice in defining non-stationarity? My hunch is that this choice might be a bit less critical than the other three above, but I still suggest checking it.

**Language and structure.** In general the English per se is on a good level. I acknowledge that every author has his/her own style of writing and way of expressing himself/herself. However, I still feel like the whole manuscript would gain a lot by paying attention to details in formulations and also if a native speaker could read through the whole manuscript in a very detailed manner to iron out circuitously and strangely formulated sentences. Time and again I came across sentences that were inaccurate and where I had a strong feeling that a native speaker would phrase it differently. Usually I understood (I think/hope) what the authors intended to say, but it didn't read smoothly.
While in some cases the style/formulation of the content certainly is debatable and also partly a matter of taste, there are so many inconsistencies and glitches in the manuscript that at some point I stopped listing them a) because I don't want the review to be longer than the manuscript itself and it should be the responsibility of the authors to read through the manuscript *before* submission and b) because in my opinion major revisions are required that demand changing most of the text anyway.
It should not be necessary to mention, but before submitting a manuscript to a journal, the author team should take care to ensure that it meets reasonable quality standards – which regrettably was not the case here. I'm not referring to the scientific content, but to the text and figures that contain reams of inconsistencies and mistakes like for example missing and erroneous axis labels, missing panel labels ("(a)"-"(f)" in Figure 2), inconsistent font size (Figure 1), inconsistent spelling of words (skilful – skillful, grid point – gridpoint, nonstationary – non-stationary), inconsistency with abbreviations (e.g., the authors use Southern Hemisphere several time before introducing the abbreviation and when they introduce the abbreviation they do it four (!) times but thereafter keep using the spelled out version) etc. etc.

Please also carefully select and structure the content of the different chapters. E.g., in the methods section there is a whole paragraph I have the impression does not belong there (L107-127), or Figure 1 shows up as a reference/result in the introduction where the work of others is reviewed, or as a further example, the definition of what the authors call a "skilful reconstruction" clearly belongs to the methods section.

For my taste, it is a rather long manuscript with many figures (11 in total). I have the impression it could be streamlined and some of the less relevant figures and content moved to a supplementary material file that goes along with the manuscript. As an example I think the whole page 9 could be condensed to 2-3 sentences, moved to a supplement or removed completely. The content presented here is neither novel nor surprising/unexpected but simply statistically obvious, well-known and does not require any sort of "analysis".

**Content.**

3.1 Similar to the example of page 9 I just made above, the whole manuscript should be streamlined and condensed to present only the essential parts which will then make enough space to address the following concerns and include the suggestions.

3.2 The authors claim in the abstract L7-8 "Non-stationarity of proxy-SAM teleconnections, as defined here, plays a negligible role in reconstructions, …", they say on L288 "To better illustrate the impact of non-stationary proxies on reconstructions, Figure 7 shows …" and also in the Discussion/Conclusion chapter (L336-338) it reads "In this study we … examine … whether or not non-stationarities in these proxy networks significantly impact the reconstructions." These statements are simply not true. I can't find any place where the authors show or analyse how these non-stationarities actually affect reconstructions. But these left-out analyses would exactly be the sort of results that would very much help making the study more valuable. E.g., I would move Fig. 7 to a supplement. What would be of interest here is how relevant non-stationarities in proxy records are for reconstructions. What is the relation to skill and how do reconstructions with a high number of non-stationary proxy records look like in comparison to reconstructions where the number non-stationary records is much lower? What is the proportion of non-stationary proxy records where non-stationarities actually become problematic for reconstructions? Just providing the chance with which a certain proportion of non-stationary SAM records will go into the reconstruction does not say anything about the effect non-stationary records have on the resulting reconstruction and its skill.

3.3 The authors sample pseudo-proxy records from random locations on the Southern Hemisphere's land mass. The results obtained with these random pseudo-proxies do not have much relevance for statements/claims/recommendations they wish to make for SAM reconstructions based on real palaeoclimate proxy records. To increase the study's relevance, I see no way around to also use the locations where we have real-world proxy records. Otherwise this study is at risk to become a pure exercise in statistics who's results are only valid and relevant for the very specific climate model that was chosen and does not provide insights that could be transferred into a broader context.

Including analyses with locations of real-world proxies will also help making what is describe on L234-235 a valid attempt, provided that the model(s) used is(are) a good enough representation of reality at these locations. At the moment I don't believe that the analyses in which Antarctica is excluded have much informative value for tree-ring-only reconstructions because a) the proxy locations are chosen randomly (on land) where mostly no real-world proxy records are available and b) for large parts the model struggles to even get the sign of the correlation right (cf. point 1.1 above and Figure 1 in the manuscript)

3.4   In the Discussion and Conclusions section (L373-377) the authors claim that the three listed points (which are very obvious and not really helpful) reduce the extent to which ENSO impacts their reconstructions. However, as under point 3.2, they don't show anywhere in the manuscript whether at all and if so to what extent these points affect their reconstructions. Analysing this possible impact is what would be of interested here. For this I suggest e.g. comparing reconstructions with randomly sampled proxy records with reconstructions that only include "good" records (according to points 1)-3) ). This would then allow the authors to make a statement in the direction they aim for here, but again with the caveat that the results may only be valid for the specific model in case the agreement of the model with reality is not sufficiently good.

Also, I wonder why the stated points should be different for seasonal data (L376-377). I think also for seasonal data it is obvious that proxy records with a significant correlation to ENSO can negatively impact the reliability of the resulting reconstructions or that proxy records with a strong enough correlation to SAM should be used.

3.5   A further point I wonder about is how much the results relating to whether SAT or precipitation record-based reconstructions perform better (in terms of the currently used skill measure) go beyond what can simply be expected statistically. The authors select records based on their correlation with SAM over the calibration period. Then a skill measure that is solely based on the correlation of the reconstruction with SAM is used. Wouldn't the result whether SAT- or precipitation-based SAM reconstructions have higher skill then simply depend on the distribution of the correlations of SAM with SAT/precipitation in the model, that is, wouldn't the reconstructions based on the climate variable with a higher proportion of correlations (absolute values) that are bigger or equal than 0.3 necessarily yield higher skill values? Or in other words, if in the model you have a higher proportion of correlation above 0.3 of SAT or precipitation with SAM, then it would be much more likely by random selection to catch pseudoproxy records that correlate strongly with SAM and hence much more likely to obtain a "skilful" reconstruction.

In conclusion, the results would exclusively depend on how the correlations in the model between SAT/precipitation and SAM are distributed (not spatially) and you don't even need to do reconstructions to know the outcome/answer.

**Minor points**

L7: I think it is not necessary to introduce the abbreviation SAT here (it is done on L73).
L7-9: Unclear sentence. "range in reconstruction skill": Does that range in any sense relate to non-stationarity? Where is that range obtained from? If it relates to non-stationarity the sentence does not make sense, because in the first part of it you say that non-stationarity plays a negligible role in reconstructions (where skill also belongs to).
L13: I would remove "nominally" here.
Figure 1 and caption:
7.1   GFDL CM2.1 and ERA not defined yet (and I think it is not done anywhere else).
7.2   Please be consistent in how you refer to the panels (a)-(d). Here for example you use (a), (c), b), d), b and d.
7.3   Very minor detail: Consider switching (a) and (c) with (b) and (d) so that in the text there reference to (a) and (c) comes before (b) and (d).
7.4   One colour bar and one x-axis label would be sufficient I think.
7.5   Different font sizes on the colour bar are used (1 and 0 is significantly larger than -1, -0.5 and 0.5). Same for the y-axes (S from 30°S on is larger than the rest).  In addition, the number next to the colour bar are extremely close to it. Add some space, because one is tempted to read -0.5 on the red half of the colour bar.
Please pay attention to such details!

7.6 Why are contours at r ≥ |0.3| of relevance here? This seems arbitrary.

7.7 I don't think a the reference for ERA-Interim is needed here (you provide it on L97).

L15: "… have been linked …": Here results from the literature are presented, but your own Figure 1 is added as reference. You should clearly separate your own results from other people's work. I think I understand the intention here (which I guess was to refer to Figure 1 as being in line with results from the literature) but this needs to be rephrased (and I'm not sure I would include your own result in the introduction here, but this is debatable).

L22-24: Ok, but feels like a bit out of place and irrelevant here. I would remove this.

L26-27: If I remember correctly there is also some work by Dave Thompson that might be relevant here and could possibly be cited. Please check and add if you feel like (and if you don't want to include it, it could at least be interesting to read :) ).

L28: "as derived" sounds strange to me.

L29: "long-term" missing before "context"?

L29: Unclear. Do you mean the past five decades by "present day"? Please be specific.

L29-30: I think the structure of this sentence does not work. I have the impression that by "present day changes" you mean the same as by "observed multi-decadal trends". Maybe just place a dot after "variability" (or otherwise rephrase).

L30: "… this, this …". Not very elegant…

L30-32: Does all this really follow from the previous sentence?? I'd remove this sentence.

L35: I don't think "meanwhile" is the appropriate word here.

L39: "can be made by examining changes" sounds strange / weird formulation…

L41: "… sensitive to *both* precipitation *or* surface air temperature …" you can't use "both" and then "or".

L42: Why is SAT not define here but only later (L73 or so)?

L42: "For example …". I feel like this sentence does not belong here. You already discussed this before.

L49: Is particle dust really relevant here and for SAM? Seems to be out of place. I'd remove this.

L55: Not sure about the formating here. Think I would write "… (CPS; Abram et al. …)".

L62: No comma after "regional".

L64: "comparing" instead of "compare".

L74: "*a* significant positive correlation" or "significant positive correlation*s*".

L78-81: Isn't this true also for longer calibration windows?

L82: Do you really "quantify" the "uncertainties" mentioned?

I noticed a mix of past and present tense in the methods section. I think generally only one should be chosen and then used consistently.

L89: Using "is" here sounds strange ("The data … is …").

L94: ERA-40 does not appear in Figure 1b and d, but the way you refer to Figure 1 it should.

L97: You miss saying that the reanalysis data are correlated with the Marshall SAM index.

L100: What do you mean by "transitions from its positive to its negative node".

L104-105: Yes, but the present study is about annual reconstructions and later on in Section 4 it is mentioned that CMIP5 generation models have issues in representing SAM-SAT relationships on a seasonal scale. Hence, saying that Gallant et al. also use CM2.1 due to its seasonal skill (austral winter) is a very incomprehensible argument for using CM2.1 (and why is "austral winter" capitalised?)

L110: "also"?

L111: "also"?

L116-117: Strange sentence.

    37.1  I think it should read "*By* using a model framework …".

    37.2  "robustly". Really?

    37.3  "windows in time" → "time windows".

    37.4  "the skill of index reconstruction" sounds weird.

L107-127: This text seems not to belong to the "Methods" section.

L120: I don't think "Alternatively" is the right word here.

L122: What do "dust particle records" do here? I'd remove this sentence (and also it seems strange to start it with "Finally" and then you begin the next sentence with "In addition").

L123: "In addition to *this*, …" "this" is very unspecific here. Please rephrase. Also, I've notice you use "this" at various places in a similar unspecific way. Please check and clarify where necessary.

L125: "… during an entirely different season." instead of "… during a different season entirely."?

L127: Consistency with what? Why should annual means be more consistent than seasonal analyses? I'm not convinced by this argument.

L130: As far as I know, most commonly latitudes between 40°S and 65°S (rather than 60°S) are used for the definition of SAM and only a minority of studies use a different definition. Is there any reason that would speak for 60°S instead of 65°S?

L134: I don't think "established" is suitable here. Maybe "modelled" or something in that direction? Also I'd use plural ("running correlations").

L132 (and whole manuscript): Be concise and consistent throughout the manuscript when using "proxy", "proxy record", "proxy data" and "proxy archive". Here I would maybe use "proxy record".

L132 (and whole manuscript): I just noticed here that you use "the SAM" and three lines later only "SAM". Please be consistent.

L135 (and whole manuscript): Here you write "non-stationary" and on the next line "nonstationary". Please be consistent.

L147: I'd replace "-" with a comma and on the next line there is a "the" missing before "SAM index".

L148-149: I don't think this sentence is correct. You say the red noise is a combination of random Gaussian noise and lag-1 autocorrelation of a climate variable time-series. But random Gaussian noise is a time-series, whereas the autocorrelation of a time-series is a number. The formula is correct, but the error happened in the attempt to put it into words I think.

L152-154 sounds a bit strange / weird formulation. Please rephrase.

L165: "metric"? Do you mean "climate variable". Also "deemed" sounds strange to me.

L169: "For this reason" does not sound logic here.

L182: Strange sentence. Is there a word missing?

L185-188: I'm happy with the choice of CPS as reconstruction method, but I think you can't say in such a general sense that it is considered to be superior to other methods.

L192: Where do you define how your model "nino3.4" index is calculated. I think this should be mentioned somewhere.

L194: Where do you introduce the abbreviation "n3.4"?

At this point I will stop pointing to typos, inconsistencies and unclear sentences, because, as explained above, I think major revisions of the figures and text is necessary anyway.

Maybe a single last comment to Figure 3, its caption and the use of "teleconnection":
    - Do you really to display the legend four times?
    - The sentence that starts with "This is therefore …" does not follow from the previous. Also I don't understand why you say "running" windows.
    - Sometimes you use "teleconnection" as substitute for "(running) correlation". These are generally not necessarily the same. While teleconnections are more "general", correlations may only capture a specific aspect of a teleconnection.

---

## Author Comment (AC1) · 22 Feb 2021

The comment was uploaded in the form of a supplement:
https://cp.copernicus.org/preprints/cp-2020-133/cp-2020-133-AC1-supplement.pdf

---

## Author Comment (AC2) · 22 Feb 2021

The comment was uploaded in the form of a supplement:
https://cp.copernicus.org/preprints/cp-2020-133/cp-2020-133-AC2-supplement.pdf
* * *

---

## Author Response (AR1)

Response to Reviewer 1

We thank the reviewer for their constructive feedback and thoughtful suggestions. Below, we address each comment with the reviewers text in bold, and our responses following. Excerpts from the text are presented in italics and additions are in red.

**2.0. Abstract**
**• The authors should mention the complete acronyme of the model, i.e. GFDL CM2.1, at least in the abstract. More generally, the authors go back and forth between "CM2.1" and "GFDL CM2.1" in the main text. Two options:**

**1. They should mention it once in the main text followed by brackets mentioning the short appellation : "[...] GFDL CM2.1 (CM2.1 hereafter) [...].**
**2. Otherwise they should call it with the long name in the whole text and figures. Since the model is called "CM2.1" in figures, option 1 seems the less constraining.**

Thank you for spotting this. We will note the model's full name in the Abstract and follow the convention of 'option 1' for the remainder of the manuscript.

**2.1. Introduction**
**• Figure 1: Maps are only partially informative since no confidence levels for correlations are shown. Instead, the authors use contours for |r| > 0.3, and I guess the other levels of contours are drawn every 0.1 steps (should have been mentioned in caption if that's the case). I understand these contours are useful when using continuous colorscales, I personally don't like it but that's not an issue here. However, the authors should at least add on maps which correlations are significant, and which are not (at the 90% confidence level, for instance). Indeed, for model simulations and instrumental data used here, time frames over which correlations are calculated have very different lengths (500 and 36 respectively), such that 0.3 correlation (for example) yields very different levels of significance for the two cases. For instance, the Student test from McCarthy et al. (2015) with corrected degrees of freedom can be used (see their Methods section): https://www.nature.com/articles/nature14491. I understand the authors are not showing significance levels to facilitate the comparison between observed and simulated patterns of SAM, which are indeed well matching, but the significance levels might hide this. The authors should at least address 1), and i suggest that using 2) might give a better figure, but i let the latter up to the authors:**

**1. The authors should keep colors and just indicate (for example) with little crosses the grid points for which correlations are not significant and explain that the 36 years time frame is too short to capture statistical significance, so we still see that GFDL CM2.1 seems adequate for testing SAM reconstructions. (See attached Fig. 1: same as Fig. 1b from the manuscript but with little black dots indicating correlations unilaterally not significant at the 90% confidence level, using Student test from McCarthy et al. 2015, see link in above paragraph).**

**2. The historical significance levels can probably be "boosted" using longer temperature and precipitation products, that will enable to use 21 more common years with SAM observations (1958 to 1978) and give more robust estimations of the SAM precipitation/temperature pattern. There is, for instance, the CRU TS dataset (Harris et al. 2020, https://www.nature.com/articles/s41597-020-0453-3).**

We appreciate the reviewers thoughtful suggestions for improving this figure, and this was a criticism shared by reviewer two. We have decided to replace Figure 1 with a new figure that, rather than comparing the correlations over 500 years vs. 36 years, identifies whether or not the values calculated for the ERA-Interim/Marshall SAM data fall within the range of the 36 year running correlations in the model data. It can be found below:

[Figure]

Correlations of annual-mean (Jan-Dec) SAT (a) and precipitation (b) from the GFDL CM2.1 model with the model-derived SAM,calculated over 500 years. Black dots show where the correlation of the ERA-Interim reanalysis product with the Marshall SAM index,calculated over a 36 year period from 1979-2014, does not fall within the range of the model's 36 year running correlation at each grid cell.

**• L.32: Typo. Space between "cycle" and ".".**

This has been corrected.

**2.2. Methods**
**• L89-91: This is one of the major limitations of the study I think. The authors are working on a control run while using a correlation-based weighting technique (weighted CPS). In**

**the real world, the observed SAM has a trend, which is likely to be due to human activity as mentioned by the authors in the introduction. Thus, since most of the temperature signals also has trends in the real world, one can expect calculating correlations (and then weights here) overestimated for the proxies that has the closer trend to the SAM one. In other words, If this study aims at being a support for real-world SAM reconstructions, the authors should mention as a caveat the fact that contrary to the pseudo proxy experiment, the correlations calculated in a real-world experiment over calibration periods might be artificially overestimated/underestimated because of the specific SAM trend.**

We strongly agree with the reviewer on this point and have made additions to the discussion to emphasise this. While this does represent a caveat, our results show that even in a 'stable' climate, significant uncertainty in SAM reconstructions can occur, and we would expect this to simply be exacerbated by calibration over the modern, anthropogenic trend.

The new discussion paragraph reads as follows:

*"A caveat of this study is our use of annual mean data, rather than seasonal fields. This is a distinction from previous real-world reconstructions utilising tree ring records (Zhang et al., 2010; Villalba et al., 2012; Abram et al., 2014; Dätwyler et al., 2018) which are not only more sensitive to SAT or precipitation of a particular season, but also combine these with other proxies such as ice cores (Zhang et al., 2010; Abram et al., 2014; Dätwyler et al., 2018), corals (Zhang et al., 2010) and lake sediments (Abram et al., 2014; Dätwyler et al., 2018) each of which may be more or less seasonally sensitive to multiple  climatological fields. In addition, many proxies such as tree rings (Cullen and Grierson, 2009; Villalba et al., 2012) have been shown to have a lag relationship with SAM from the previous year, which is also not accounted for in this study. Dätwyler et al.(2020)'s 'perfect' pseudoproxy experiments for an austral summer SAM show similar reconstruction skill to our results (an average 31yr running correlation of ~0.7-0.8 for their ensemble mean) and while the methods of this study are not analogous to theirs, it supports the conclusion that proxy-derived reconstructions of the SAM in a model framework can, at best, reproduce  50-60% of SAM variance on an annual time-scale.*
*Finally, the results we present here are derived from a control simulation and the uncertainties in our reconstructions represent noise internal to the climate system. This is in direct contrast to real-world reconstructions which have the bad fortune of requiring proxies to be calibrated over a period with a significant anthropologically forced trend in the SAM. We would expect this to increase the uncertainty in reconstructions and any future model-based verification of real-world reconstructions would need to address this problem."*

**• L95: "CM2.1 was selected due to its relatively good representation of the SAM.". Since this is the major reason for which the authors use this model, we need an objective statistical test that shows that spatial correlations of SAM with temperature and precipitation simulated from CM2.1 are significantly similar to those from the observations at a given significance level. This is far more convincing than just "CM2.1 has a relatively good representation" with a subjective use of "relatively". I guess a**

**Monte-Carlo approach using distance between pairs of random correlation maps can do the job. I have no doubts that the statistical test won't affect the model choice of the authors.**

We believe that the inclusion of the new Figure 1 sufficiently addresses this point, as well as the addition of reference to the Bracegirdle et al. 2020 study, which highlights CM2.1's ability to simulate the position and strength of the southern hemisphere jet favourably when compared to the CMIP5 and CMIP6 suite of models. This is of course in addition to the references already included in the manuscript which have performed far more thorough analyses on the model's performance. We have revised/rephrased the methods section, and it now reads as follows:

*"CM2.1 is selected due to its good representation of the SAM compared to similar models from the CMIP5 and CMIP6 archives (Bracegirdle et al., 2020) while Karpechko et al. (2009) find performance to be favourable when compared to ERA-40 data. The spatial structure of the SAM is well simulated, accurately capturing the centre of action over the Pacific, while being slightly too zonally symmetric on the eastern half of the Southern Hemisphere (Raphael and Holland, 2006). Importantly for our purposes, CM2.1 accurately simulates the latitude at which the SAM transitions from its positive to its negative phase (as expressed via regression onto 850hPa winds) over South America, which many models of a similar age and computational complexity fail to achieve (Raphael and Holland (2006), their Figure 4b). The amplitude of the model SAM index is comparable with observations (Raphael and Holland, 2006), although its variability is larger than observed (Karpechko et al., 2009). As previously noted, we should be cautious directly comparing observations (in this instance, ERA-Interim (Dee et al., 2011) data, correlated with the Marshall SAM index (Marshall, 2003) over the 36 year period from 1979-2014) spanning a brief time period with our model data which spans 500 years. To address this, we calculate a 36 year running correlation between the model SAM and our SAT and precipitation fields and identify if the correlations derived from observations fall within the model range (Figure 1). The SAM index in the model is calculated according to the method of Gallant et al. (2013) as the difference in normalized, zonally averaged sea level pressure anomalies between $40 \circ S$ and $60 \circ S$. Aside from a region in equatorial South America in the SAT field, the agreement is good, with 87% of SAT and 95% of precipitation grid cells on land showing agreement with observations."*

• **L144: I guess the authors mean "$a_0$ and $a_1$ are regression coefficients [...]", instead of "$a_0 + a_1$ are regression coefficients [...]"**

The reviewer is correct. This has been amended in the revised manuscript.

• **L167: I finally understand the relevance of contours from Fig. 1. As for the comment in the introduction, the choose of |r| > 0.3 is here purely arbitrary. Would we obtain the same results with other arbitrary choices like 0.25 or 0.4? I think a way for the authors to gain in transparency in their methodological set-up is to make a selection of proxies where the correlation significance at a given level is used for selecting them, instead of using the absolute values of correlations, that have different meanings for 31, 61 and**

**91 window length. I know the authors have been honest and claimed that this choice is arbitrary, but I am wondering if a (pesudo)proxy selection based on a statistical test is also geographically too restrictive. If not, I think the authors should consider this way for selecting (pesudo)proxies.**

We thank the reviewer for their considered comments on this issue. Our choice to use a correlation criteria finds precedence in McGregor et al. 2013 (doi:10.5194/cp-9-2269-2013) and Batehup et al. 2015 (doi:10.5194/cp-11-1733-2015) and the motivation is twofold: firstly, to set a reasonable standard for proxy skill to ensure they actually capture the SAM signal to some degree. Secondly, it highlights the result we present in Figure 2: that a 'good' correlation (of 0.3, for example) over a shorter window may not represent the proxies' true correlation over 500 years and, when using a method such as an *r*-weighted CPS, this assumption should result in far more uncertainty in the range of reconstruction skill for these shorter windows. This would not be as clearly visible for some significance criteria, as a longer window results in poorer correlation coefficients meeting more stringent significance criteria (see below figure).

We would also add that the choice of 0.3 (as opposed to 0.25 or 0.4 as suggested) as a cutoff value would be similar to the choice required for a significance criteria, in other words do we select the cutoff at p < 0.1, 0.05 or 0.01? Each would similarly change the number of proxies available for selection and by extension, the resulting reconstructions.

[Figure]

This figure shows the running correlation values for every land cell plotted against the significance of the correlation (calculated using the Ebisuzaki method outlined in Abram et al. 2014). Panels on the top row are for SAT, while those on the bottom are for precipitation.

We have updated the methods section to include references presented above. It now reads:

*"Proxies are randomly selected in accordance with several conditions. The proxy must be on land in the Southern Hemisphere and must have a correlation with the model SAM index of |0.3| or greater within the calibration window after the method of McGregor et al. (2013) and Batehup et al. (2015). While a correlation of 0.3 is arbitrary in choice, it ensures that the proxy represents the SAM to some extent while not being so high that proxies are only sourced from a geographically limited region."*

In addition the discussion and conclusions section has been updated to read:

*"Increasing the calibration window does not increase the chance of producing a more skilful reconstruction, it does however,along with maximising the number of proxies, cause the range of reconstruction skill to converge on the skill of our 'perfectly' calibrated proxy reconstructions (blue envelopes, Figures 4 and 5). It should be noted that this will be, in part, due to our correlation requirement of r >= |0.3| for proxies imposing a progressively more rigorous selection criteria for longer calibration windows. Adding more sites to a reconstruction has limited benefit in terms of the maximum skill it can achieve, with values largely plateauing at a network size of~20. When it comes to minimum skill, however, this improves for increases in network sizeall the way up to and including network sizes of 70 proxies (Figure 3). This in turn, acts to increase the proportion of skilful reconstructions for a given window size."*

**• L188: This statement is way too strong. Probably that in VonStorch's 2009 study, weighted CPS is better in skill than the PCR method. However that might not be true for other datasets (maybe for the author's dataset who knows). There is in-deed a very famous mathematical theorem, named the "No free lunch" theorem, that stipulates that given two statistical methods A and B, we can always find datasets for which A is better than B, and others for which B is better than A (Wolpert 1996, https://www.mitpressjournals.org/doi/abs/10.1162/neco.1996.8.7.1341). With this in mind, the authors should remove this statement and just say that the reconstruction sensitivity to the method is beyond the scope of the study.**

We concur with the reviewer - this statement is indeed too strong, and has been removed as requested.

**• L190-191: Are the authors calculating the skill of the correlation using the 500 years of the control simulation? If yes, this is problematic and it should be fixed, because this means that skill scores for longer calibration time frames will be artificially increased, only by the fact that they are including more data over which the statistical model is built. If not, this should be clarified in the main text because this point is not clear to me.**

The reviewer is correct in their assessment - the reconstructions are validated over the full 500 years of the data (including the calibration period) as is done in Batehup *et al.* 2015. The reviewer is correct that, for real world reconstructions, we would validate over a separate period to the calibration window to ascertain an unbiased estimate of the reconstruction skill that then presumably represents its skill over the full time-interval of the data. In this instance, we have no interest in this as we can simply validate each reconstruction over the full 500 year time-period (and therefore also allowing a direct comparison to our 'perfect' or 'true' 500 year reconstructions, which by their nature require validation over their calibration period).

The effect of this, rather than artificially increasing the skill of the reconstruction, is to increase its convergence with the 'true', 500 year calibrated reconstructions we create (visible in Figures 4 and 5). For example, we can imagine reconstructions calibrated over a very long window of 470 years, then validated over only 30 years, the resulting spread of calculated 'skill' would be larger than in actuality.

We agree with the reviewer that this was not stated clearly enough in the methods section and have amended the manuscript as follows:

*"To quantify the skill of the pseudoproxy reconstructions, Pearson correlation coefficients are calculated between each SAT/precipitation-derived SAM index and the sea level pressure-derived SAM index over the full 500 years of data."*

**2.3. Results**
**• L198: Please address my last statement from the Methods section because this might affect the first statement of this section.**

The results in Figure 2 are independent of the concern the reviewer raises in their comment regarding the validation of reconstructions over the full 500 years of data. This figure merely displays, for all land points, the 'apparent' vs. 'true' correlation for a pseudoproxy.

**• L211-212: Seems a bit arbitrary too to say that a reconstruction is skillful when r 2 > 0.5, although this is a relatively constraining criterion. I do not require changes here, but the authors should keep in mind that using more relevant evaluation metrics such as the coefficient of efficiency or the reduction of error (discussed for instance in Macias-Fauria et al. 2012, https://www.glaciology.net/pdf/Macias-Fauria-Dendrochronologia12-P ersistence.pdf ) can clearly help for addressing this issue (and probably the last one too).**

We thank the reviewer for raising this concern (shared by reviewer 2). We believe our arbitrary 'skill' criteria to be a relatively generous one, selected primarily as a way to summarise the thousands of reconstructions across our parameter space using some cutoff criteria (in this instance - can the reconstruction capture at least half the variance of the true model SAM?). We acknowledge, however, that a secondary measure would complement our approach. To that end, we will include the following figure in the Appendix. It displays the median root mean

square error across the 10,000 reconstructions calculated for each network size. An additional point of discussion has been added to the manuscript which reads:

*"Assessing the skilfulness of our reconstructions, where skilfulness is defined as being able to reproduce ≥ 50% of SAM variance over the full 500 years, reconstructions derived from precipitation performed best (Figure 3), with a maximum of 91% of reconstructions being reported as skilful (91 year window, 70 proxies) as well as less spread due to variability of the teleconnection between precipitation and the SAM. SAT-derived reconstructions performed poorly by comparison, with only 25% qualifying as skilful (61 year window, 70 proxies). It is worth noting that this result remains consistent when examining a different measure for skill. If we consider median root mean square error (RMSE), precipitation derived reconstructions perform better and aswith our threshold skill score, the RMSE improves as we increase the number of proxies in a reconstruction (Figure A2)."*

[Figure]

This figure shows median RMSE across the 10,000 reconstructions calculated for each network size for SAT (a,b,c) and precipitation (d,e,f). Panels a) and d) show results for the 31yr calibration window while panels b) and e) show results for the 61yr window and c) and f) show errors for the 91yr window.

**• L230: Is there any reference for this so-called "perfect" reconstruction approach for calculating skills? I feel like this is an "handmade" skill metric, but please let me know if I**

**am wrong. If I am wrong, the authors should add a reference then. Naively, i would say that the "perfect" reconstruction has 1 correlation with the pressure-based model SAM. Here, the authors' definition of perfect is purely relative to the time period covered by the control simulation. If we add 200 more years of simulation, is the 700-years based pseudoproxy reconstruction "more perfect" than the 500-years one? Considering the last two comments about the metric used and correlation calculated over the 500 years (including calibration), I think that I see why the authors use this "perfect reconstruction" skill metric. Here again, I think using more relevant metrics than correlation could avoid overfitting, handmade metrics, and some arbitrary choices.**

We apologise to the reviewer that our approach was not sufficiently clear here. When we refer to a 'perfect' reconstruction, it is not one that has an r = 1 with the model SAM, but rather one where the proxies are calibrated over the entire reconstruction period (500 years in this instance). This is not intended to be a measure of skill, but rather to act as a point of comparison, displaying the limits of our reconstruction approach in the model framework we have selected. To avoid this confusion, we have changed the use of the term 'perfect' to a more appropriate 'true' reconstruction throughout the manuscript.

**• L284: It's good to know that non-stationary relationships tend to fall in regions for which correlations are weak. It would have been even more interesting to see that these correlations are not significant (cf. my comments on Fig 1).**

As the reviewer suggests, we have calculated the significance of the 500 yr correlations using the Ebisuzaki method described above (see figure below). We find that despite these weak correlation coefficients, virtually all are significant at p < 0.1 (to be specific, all correlations greater than 0.08). Below we present a figure visualising this. In addition, we have noted this in the manuscript as suggested. This sentence now reads:
*"is also noteworthy that these non-stationary regions (as calculated using the 31 year running correlation) appear to fall, on average, in regions where correlations are weaker (though still significant at p< 0.1 when r>0.08) over the full 500 years (Figure 1a and b). "*

[Figure]

[Figure]

**• Figure 10: The authors say hatched points are for p > 0.5. I guess they mean p < 0.05. Otherwise it has no sense.**

We thank the reviewer for spotting this mistake. It has been corrected.

**2.4. Discussions and conclusions**
**• First two paragraphs: The authors claim that reconstructions derived from temperatures are more skillful. But with real world data, we often use both precipitation and temperature proxies. Is there any reason for not considering the case of a mixing of both types of data? Should at least been mentioned in the discussion.**

A third set of reconstructions utilising both types of proxies was not pursued as the parameter space being covered in this study is already considerable. While this issue of differences with real-world reconstructions was addressed to an extent later in the discussion, we have made an addition to the following paragraph (new text in read) to more explicitly highlight this.

*"A caveat of this study is our use of annual mean data, rather than seasonal fields. This is a distinction from previous real-world reconstructions utilising tree ring records (Zhang et al., 2010; Villalba et al., 2012; Abram et al., 2014; Dätwyler et al.,2018) which are not only more sensitive to SAT or precipitation of a particular season, but also combine these with other proxies such as ice cores (Zhang et al., 2010; Abram et al., 2014; Dätwyler et al., 2018), corals (Zhang et al., 2010) and lake sediments (Abram et al., 2014; Dätwyler et al., 2018) each of which may be more or less seasonally sensitive to multiple climatological fields."*

**• L343-344: Why comparing the proportion of skillful reconstructions of SAT and precip for 61-years and 91-years calibration window lengths, respectively?**

What we report here are the best results for the respective fields (SAT and precip) - for SAT, a calibration window of 61 yrs with 70 proxies produced the best result while precipitation-derived reconstructions produced the best result when calibrated over 91 years and with 70 proxies. We apologise to the reviewer that this was not more clearly articulated and have reformulated the text. It now reads:

*"Assessing the skilfulness of our reconstructions, where skilfulness is defined as being able to reproduce ≥ 50% of SAM variance over the full 500 years, reconstructions derived from precipitation performed best (Figure 3), with a maximum of 91% of reconstructions being reported as skilful (91 year window, 70 proxies) as well as exhibiting less spread due to variability of the teleconnection between precipitation and the SAM. SAT-derived reconstructions performed poorly by comparison, with only a maximum of 25% of reconstructions qualifying as skilful (61 year window, 70 proxies)."*

**• L374: 1) Or maybe that 31 years windows are simply too short to robustly estimate the significance of the correlation with ENSO.**

**• L374: Suggestions 1, 2 and 3 for avoiding an ENSO bias in the reconstruction should be discussed a bit more. Do they mean that we need to wait 91 years of direct SAM observations (at least year 2049 then) that overlaps with at least 70 SH proxies (so, far more than 2049) that all are not significantly correlated with ENSO but all significantly correlated with SAM? That's personally what I concluded from these suggestions.**

We respond to the above comments together, as this section has been re-written. Instead of making concrete statements regarding the requirements for a reconstruction free from ENSO bias, the aim here was to highlight that, similar to non-stationary proxies, we can improve our chances of a more skillful reconstruction by calibrating over a longer window. This is because the variance in proxy-SAM correlation decreases, meaning that even though ENSO may be responsible for a large proportion of this variance, it has less of an impact on the reconstruction. We have re-written this section and it now reads as follows:

"*It is not unreasonable to suspect that ENSO may be contributing towards the proxy-SAM teleconnection variance. Dätwyler et al. (2020) identify a highly variable, but centennial-average of -0.3 between austral summer ENSO and SAM reconstructions over the last millennium. Their pseudoproxy experiments using a CESM1 ensemble show significant changes in SAT during periods of large negative SAM-ENSO correlation (their Figure 4, bottom left panel). The pattern is similar to our results (Figure 9a), with regions of significant correlation over much of Antarctica and three regions in the Southern ocean centered on roughly 60ºE, 150ºE and 60ºW. Rather than excluding proxies whose teleconnection with SAM is significantly correlated with ENSO, we can minimise ENSO's impact simply by calibrating over a longer window thus ensuring that, while ENSO may impact these proxies, the variance of their teleconnection with SAM will be small. Its greater impact at longer windows (Figure 11) is therefore minimised as the variance of the proxy-SAM teleconnection is smaller (Figure 8).*"

Response to Christoph Dätwyler

We thank Christoph Dätwyler (C. D. hereafter) for his constructive feedback. Below, we address each comment with the reviewers text in bold, and our responses following. Excerpts from the text are presented in italics and changes from the original manuscript are in red.

**Major points**
**1 Robustness. I have major concerns regarding the robustness of the presented results. I do have the impression that the results depend too much on several choices the authors made. Generally speaking, the results must/should not depend too heavily on such choices to be of any significant value. Or if there is a strong dependence on a choice, this choice must be very well justified. Creating a supplementary material file to complement the manuscript would allow the authors to incorporate the results of robustness tests. Before the manuscript should be considered for publication, checking the robustness of the following points is irremissible.**

**1.1 How strongly are the results model-dependent? It is well-known that climate models struggle to capture high-latitude dynamics. The model they use is already about 15 years old and I wonder whether there have not been any advances since then. Using a model that is as good as possible is of vital importance for this study if any of the conclusions drawn should have a meaning in real world scenarios.**
**Moreover, the authors also (partly) justify their choice of CM2.1 with Figure 1 and state that there is a spatially good agreement between the correlations (of SAM with SAT and precipitation) in the model and the reanalysis data (L95-97). I don't agree with this statement because if we, e.g., look at SAT and exclude Antarctica then I'd estimate (visually) that about half of the grid points that are over land (i.e. South America, Australasia and Africa) do not even share the same sign. Adding data from one or more additional model, would strongly improve the quality of the paper, as it would allow to assess all conclusions with regards to the choice of model data.**

We agree that the quality of the model simulations is important to draw meaningful conclusions from the data. While it is true that CM2.1 is an older model in the CMIP5 archive, its biases in both the position and strength of the Southern Hemisphere (SH) jet are very favourable compared to both contemporary models and newer models of the CMIP6 generation (see Bracegirdle et al., 2020 - https://doi.org/10.1029/2019EA001065 ).

Regarding Figure 1 - the comparison of 36 years of data with 500 years is also a concern raised by reviewer one. This is particularly true as over the observational period there is a strong positive trend in the SAM, potentially resulting in correlations with temperature and precipitation we would not expect in the mean control state of a pre-industrial simulation. As a result we will be replacing Figure 1 in the manuscript with the first and third panels from the following figure. This figure shows the same mean 500 year correlations of the model SAM and temp/precip, but overlaid is whether the values from the ERA-Interim/Marshall SAM correlation do not occur within the range of a running 36 year correlation at each grid point. In comparison, we

performed the same analysis on the GISS-E2-1-G 850 year control simulation which, according to Bracegirdle et al. (2020), has among the smallest combined biases in the SH jet in the new CMIP6 archive. Precipitation correlation fields in both models are largely the same, whereas for temperature, there are differences in southern Africa (a weak negative correlation in CM2.1 and a weak positive correlation in the GISS model). We believe that CM2.1 is sufficient to address the aims of this study and produce meaningful results.

CM2.1 - SAT

[Figure]

GISS - SAT

CM2.1 - Precip.

GISS - Precip

**1.2 How much different would the results look if for the pseudo-proxy selection, a
different absolute value for the correlation with the model-based SAM index would be
chosen? I don't even think fixing an absolute value does make sense at all because**

**different window width for calibration are used. It is much easier to get a high correlation when correlating only over 31 years as compared to 91 years. Rather, the choice whether a record is goes into the reconstruction should be based on whether the correlation with the target is significant or not.**

C. D. suggests that rather than using a fixed correlation criteria for proxy selection, that a significance test would be more appropriate. Below we present a figure of running correlations for all land cells vs. their significance (as calculated against 1000 synthetic time series using the method of Abram et. al., 2014). If standard significance criteria ($p < 0.1$) are applied, then virtually all the proxies we select for reconstructions would be included, including those of poorer correlations for the 61 and 91 year windows (as well as the 31 year window for SAT). Applying a significance screening would simply result in the same question - how would reconstructions change if we applied a cutoff of $P < 0.05$ or $0.01$ instead of $0.1$, for example?

C. D. states that our reconstructions would be artificially more skilful for longer windows due to an r value of 0.3 being easier to achieve over 31 than 61 or 91 years. This is, however, precisely the point - as the r value over a longer window of 91 years will be more likely to resemble its 'true' r value over the full 500 years (for example, in Fig. 2 of the manuscript we can see that a cutoff of r >= 0.3 ensures that the r value calculated over the 61 and 91 yr windows must be the same sign as the 'true' correlation over 500 years), we would expect the range in the skill of reconstructions to reduce. How much it reduces is one of our key results in this instance.

We agree that this could have been articulated more clearly in the methods and have amended this paragraph in the methods, including a reference to McGregor et al. 2013 and Batehup et al., 2015 whose method we follow.

*"Proxies are randomly selected in accordance with several conditions. The proxy must be on land in the Southern Hemisphere  and must have a correlation with the model SAM index of |0.3| or greater within the calibration window after the method of McGregor et al. (2013) and Batehup et al. (2015). While a correlation of 0.3 is arbitrary in choice, it ensures that the proxy represents the SAM to some extent while not being so high that proxies are only sourced from a geographically limited region."*

In addition, we have also explicitly highlighted the impact our cutoff criteria has on the results in the discussion section:

*"Increasing the calibration window does not increase the chance of producing a more skilful reconstruction, it does however,along with maximising the number of proxies, cause the range of reconstruction skill to converge on the skill of our 'perfectly' calibrated proxy reconstructions (blue envelopes, Figures 4 and 5). It should be noted that this will be, in part, due to our correlation requirement of r >= |0.3| for proxies imposing a progressively more rigorous selection criteria for longer calibration windows. Adding more sites to a reconstruction has limited benefit in terms of the maximum skill it can achieve, with values largely plateauing at a network size of~20. When it comes to minimum skill, however, this improves for increases in network sizeall*

*the way up to and including network sizes of 70 proxies (Figure 3). This in turn, acts to increase the proportion of skilful reconstructions for a given window size.*"

[Figure]

This figure shows the running correlation values for every land cell plotted against the significance of the correlation (calculated using the method outlined in Abram et al. 2014 utilising synthetic time-series). Panels on the top row are for SAT, while those on the bottom are for precipitation.

**1.3 How much does the choice of r² ≥ 0.5 / r ≥ 0.71 when defining a "skilful reconstruction" affect the outcome? This choice is very cumbersome to me and I cannot see any justification for it. By just looking at the figures, I suspect choosing e.g. r ≥ 0.6 or r ≥ 0.8 would completely invalidate the conclusions that stand in connection with this measure of skill. Given that the authors screen their proxy location using correlations, it is to be expected that a reconstruction rated with a correlation-based skill score will perform rather well. Instead, a skill measure that is different from the condition used for screening would allow a more robust assessment. Furthermore, since this definition of skilful is purely looking at the correlation of the reconstruction with the model-derived SAM index, the reconstruction could theoretically be completely off and still have perfect skill, or the reconstruction could have lost almost all its variance and be nearly completely flat while still having an extremely high correlation.**
**There are many possibilities that could potentially help here. I am think of accuracy measures such as e.g. RMSE, RE and CE. I hope this suggestion helps the authors to find a justifiable way of measuring skill.**

We thank the C. D. for his thoughts here, as this was also raised by reviewer one. The motivation was to create an efficient summary of the reconstruction skill using some threshold criteria, i.e. this % of reconstructions can be considered skilful given some cutoff. Other accuracy measures such as RMSE, RE and CE would face a similar problem; selection of a cutoff value to define as 'skilful'. In this case, we set what we believe to be a very low bar by requiring only 50% of the variance to be reproduced by the reconstruction to be considered skilful. As the C. D. suggests, increasing this threshold would decrease the % of reconstructions defined as skilful, but the overall conclusion would be the same - more proxies, sourced from a global domain create more skilful reconstructions (as measured by the proportion of variance reconstructed).

We do however agree with the shortcomings of calculating the coefficient of determination for data that has been calibrated via correlation and that this may not show the same result as some other measure of error. As a result, a supplementary figure will be included with calculated values for median RMSE across the 10,000 reconstructions for each network size to confirm that the choice of validation statistic does not change the conclusions presented (see below). The following text has also been added to the discussion and conclusions section of the manuscript:

*"Assessing the skilfulness of our reconstructions, where skilfulness is defined as being able to reproduce ≥ 50% of SAM variance over the full 500 years, reconstructions derived from precipitation performed best (Figure 3), with a maximum of 91% of reconstructions being reported as skilful (91 year window, 70 proxies) as well as less spread due to variability of the teleconnection between precipitation and the SAM. SAT-derived reconstructions performed poorly by comparison, with only 25% qualifying as skilful (61 year window, 70 proxies). It is worth noting that this result remains consistent when examining a different measure for skill. If we consider median root mean square error (RMSE), precipitation derived reconstructions perform better and aswith our threshold skill score, the RMSE improves as we increase the number of proxies in a reconstruction (Figure A2)."*

[Figure]

Root mean square error across the 10,000 reconstructions calculated for each network size for SAT (a, b, c) and precipitation-derived reconstructions (d, e, f). Results are displayed for the 31 (a, d), 61 (b, e), and 91 (c, f) year calibration windows.

**1.4 How strongly do the results depend on the 10%-choice in defining non-stationarity? My hunch is that this choice might be a bit less critical than the other three above, but I still suggest checking it.**

It is not clear what C. D. is asking here - if it is to ask what the impact would be if we chose a less rigorous definition (5% instead of 10%) we certainly find an increase in non-stationary proxies, particularly in the 31 year window - but this is precisely why we employ the stricter 10% threshold, because 5% is what we would expect through random chance.

**2 Language and structure. In general the English per se is on a good level. I acknowledge that every author has his/her own style of writing and way of expressing himself/herself. However, I still feel like the whole manuscript would gain a lot by paying attention to details in formulations and also if a native speaker could read through the whole manuscript in a very detailed manner to iron out circuitously and strangely formulated sentences. Time and again I came across sentences that were inaccurate and where I had a strong feeling that a native speaker would phrase it differently. Usually I understood (I think/hope) what the authors intended to say, but it didn't read smoothly.**

**While in some cases the style/formulation of the content certainly is debatable and also partly a matter of taste, there are so many inconsistencies and glitches in the manuscript that at some point I stopped listing them a) because I don't want the review to be longer than the manuscript itself and it should be the responsibility of the authors to read through the manuscript before submission and b) because in my opinion major revisions are required that demand changing most of the text anyway. It should not be necessary to mention, but before submitting a manuscript to a journal, the author team should take care to ensure that it meets reasonable quality standards – which regrettably was not the case here. I'm not referring to the scientific content, but to the text and figures that contain reams of inconsistencies and mistakes like for example missing and erroneous axis labels, missing panel labels ("(a)"-"(f)" in Figure 2), inconsistent font size (Figure 1), inconsistent spelling of words (skilful – skillful, grid point – gridpoint, nonstationary – non-stationary), inconsistency with abbreviations (e.g., the authors use Southern Hemisphere several time before introducing the abbreviation and when they introduce the abbreviation they do it four (!) times but thereafter keep using the spelled outversion) etc. etc. Please also carefully select and structure the content of the different chapters. E.g., in the methods section there is a whole paragraph I have the impression does not belong there (L107-127), or Figure 1 shows up as a reference/result in the introduction where the work of others is reviewed, or as a further example, the definition of what the authors call a "skilful reconstruction" clearly belongs to the methods section.**

**For my taste, it is a rather long manuscript with many figures (11 in total). I have the impression it could be streamlined and some of the less relevant figures and content moved to a supplementary material file that goes along with the manuscript. As an example I think the whole page 9 could be condensed to 2-3 sentences, moved to a supplement or removed completely. The content presented here is neither novel nor surprising/unexpected but simply statistically obvious, well-known and does not require any sort of "analysis".**

We thank the C. D. for his thorough review of the grammatical aspects of the manuscript and we naturally regret the errors and oversights he has identified. The latest iteration has been thoroughly read-through and altered/ streamlined where appropriate.

To address other comments:
- While we regret the missing panel labels on Figure 2, this was due to a PDF processing error by the journal which we also experienced with one other figure. Ideally this will not be repeated in subsequent submissions.
- The paragraph being referenced here has been moved to a more appropriate location in the Introduction and streamlined into the existing text. The new paragraph reads as follows:

  *"This study aims to quantify the uncertainties raised by the aforementioned assumptions within a modelling framework, similar to Batehup et al. (2015), and seeks to address the following questions: 1) What impact does proxy network size and calibration window size have on the skill of a resulting reconstruction? 2) How does the geographical distribution*

*of the proxies affect reconstruction skill? 3) Are any regions in our model framework prone to producing non-stationary proxies and what could be modulating the SAM-proxy teleconnection? The use of climate models to assess the skill of paleo reconstructions provides an opportunity to investigate a 'perfect' time-series of the climate index we wish to reconstruct and the ability to reconstruct this index with fields from a model, which act as pseudo paleo-proxies. These 'perfect' proxies are free from non-climatic noise that may degenerate a teleconnection signal in a real proxy, isolating instead changes in teleconnection strength due to underlying variability in the climate. This is in contrast to 'real world' proxies which are also prone to other influences inherent with the physical/chemical/biological nature of the proxy itself. It is often assumed that these effects will be minimised by sampling proxies from a range of regions as local factors would not be expected to be correlated amongst differing locations. Additionally, model data allows us to assess multi-decadal to centennial changes in proxy-SAM teleconnection and how calibration over certain windows in time affects the skilfulness of a SAM reconstruction."*

The remainder of  the paragraph remains in the methods section as a motivation for the use of annual-mean climatological fields in this study.

- Figure 1 has been replaced with an updated figure and will not be referenced in the introduction any longer.
- The definition of a skilful reconstruction has been moved to the methods section.
- Typographical inconsistencies will be remedied.
- We disagree that the information on page 9 is 'obvious' or 'well known'. Not only does it represent the flaw in one of the key assumptions we highlight (that of teleconnection stability), it explicitly quantifies it for the case of the SAM and helps inform how and why our resulting reconstructions vary in their ability to represent the model SAM. As a result, we have left it in the manuscript, but streamlined the text to make it more concise. It now reads:

*"The importance of a long calibration window is illustrated in Figure 2. For example, a true correlation of -0.3 between precipitation and the SAM may become anything ranging from -0.65 to 0.1 when evaluated over just a 31 year segment (Figure 2d). However, as the window size increases, it is increasingly likely that the calculated correlation is representative of the true correlation.* *For example, calibration windows of 61 and 91 years ensure that our proxy's correlation with SAM is always the  same sign as over the 500 year period (Figure 2e and f).* *Also noteworthy is the considerable decrease in the maximum available number of proxies eligible for inclusion in reconstructions when calibrating with a 61 year window, rather than a 31 year window (Table1). A smaller decrease in the proxy pool is seen when lengthening the window from 61 to 91 years."*

**3 Content.**
**3.1 Similar to the example of page 9 I just made above, the whole manuscript should be streamlined and condensed to present only the essential parts which will then make**

**enough space to address the following concerns and include the suggestions.**

As stated above, we have gone through the manuscript and streamlined it where appropriate.

**3.2 The authors claim in the abstract L7-8 "Non-stationarity of proxy-SAM teleconnections, as defined here, plays a negligible role in reconstructions, ...", they say on L288 "To better illustrate the impact of non-stationary proxies on reconstructions, Figure 7 shows ..." and also in the Discussion/Conclusion chapter (L336-338) it reads "In this study we ... examine ... whether or not non-stationarities in these proxy networks significantly impact the reconstructions." These statements are simply not true. I can't find any place where the authors show or analyse how these non-stationarities actually affect reconstructions. But these left-out analyses would exactly be the sort of results that would very much help making the study more valuable. E.g., I would move Fig. 7 to a supplement. What would be of interest here is how relevant non-stationarities in proxy records are for reconstructions. What is the relation to skill and how do reconstructions with a high number of non-stationary proxy records look like in comparison to reconstructions where the number non-stationary records is much lower? What is the proportion of non-stationary proxy records where non-stationarities actually become problematic for reconstructions? Just providing the chance with which a certain proportion of non-stationary SAM records will go into the reconstruction does not say anything about the effect non-stationary records have on the resulting reconstruction and its skill.**

We concur with C.D. here. As a result, we have moved Figure 7 to the appendix and have replaced it with the figure presented below. Changes have been made to the results section of the manuscript to reflect the inclusion of this new figure and to streamline the text. It now reads as follows:

*"To better illustrate the impact of non-stationary proxies on reconstructions, Figure 7(a) compares the skill of our SH reconstructions with the percentage of non-stationary proxies in each. The effect of non-stationary sites is negative in all but one instance. Correlations are typically stable with network size and are relatively weak, with mean $r^2$ values of 0.03. Reconstructions calibrated with a 31 year window are outliers, both of which see a slight increase in skill with larger network sizes. In particular, the positive relationship observed for the precipitation reconstructions (Figure 7a and b, purple line) suggests that these proxies provide a net benefit to the reconstructions they are part of, despite their non-stationary nature. SAT reconstructions calibrated over 61 and 91 years are noteworthy as the impact of non-stationary sites is larger ($r^2$ = 0.19 for 70 proxies calibrated over 91 years) and increases with network size, when compared to other scenarios (Figure 7a and b, yellow line)."*

[Figure]

Correlation (y-axis) between the skill of a given reconstruction and the percentage of non-stationary proxies it contains (a), plotted as a function of network size. Same as panel (a), but y-axis shows regression slope. Calculations are over 10,000 reconstructions for each network size. r = 0 is plotted as a black dashed line. All correlations are significant to at least p < 0.05 other than in the region bounded by the two red lines about r = 0.

**3.3 The authors sample pseudo-proxy records from random locations on the Southern Hemisphere's land mass. The results obtained with these random pseudo-proxies do not have much relevance for statements/claims/recommendations they wish to make for SAM reconstructions based on real palaeoclimate proxy records. To increase the study's relevance, I see no way around to also use the locations where we have real-world proxy records. Otherwise this study is at risk to become a pure exercise in statistics who's results are only valid and relevant for the very specific climate model that was chosen and does not provide insights that could be transferred into a broader context.**

We disagree that the random sampling of sites from the Southern Hemisphere (including sites that would be utilised in real world reconstructions) has no bearing on the claims we are trying to make as a broad-scale look at the parameter space affecting SAM reconstruction skill. We argue, and show, that even under the most ideal circumstances (perfect model data with a large, hemisphere-wide proxy pool and long calibration windows), 1) Variance in proxy-SAM teleconnection has a non-trivial impact on resulting reconstructions; 2) there are limits to SAM reconstruction skill that cannot be identified by correlation over a short validation period, as in real world reconstructions; 3) Reconstructions are better when proxies are sourced from multiple regions, helping to cancel our regional climatic noise.

The creation of reconstructions based upon real-world proxies is beyond the scope of this study and would require a more considered approach including the use of appropriate time-averaging of the data for each proxy (spring/summer temperatures for tree rings, for example), the consideration of lag correlations between a proxy and season (again, important for tree rings), the appropriate addition of noise to the records as well as the adjustment of record length (these records are not uniformly 500 years in length like our model data, with many being either considerably shorter or considerably longer - something we cannot achieve with this data), among other considerations.

**Including analyses with locations of real-world proxies will also help making what is describe on L234-235 a valid attempt, provided that the model(s) used is(are) a good enough representation of reality at these locations. At the moment I don't believe that the analyses in which Antarctica is excluded have much informative value for tree-ring-only reconstructions because a) the proxy locations are chosen randomly (on land) where mostly no real-world proxy records are available and b) for large parts the model struggles to even get the sign of the correlation right (cf. point 1.1 above and Figure 1 in the manuscript)**

As we have already addressed the skill of the model and the issue of randomly selected points above, we simply address the issue of a comparison to 'tree ring only' reconstructions.

We agree that the direct comparison to a 'tree ring only' reconstruction is not helpful, not least because other land-based proxy records can contribute to reconstructions (pollen records in terrestrial sediment cores, for example). As Antarctic sites constitute a large percentage of our available grid points, this comparison of with-and-without Antarctica was in part to ensure that it was not disproportionately impacting the skill of our reconstructions. The manuscript has been updated and now reads:

*"As Antarctica represents a large percentage of the available proxies (Table 1), reconstructions are included for proxies sourced from the entire Southern Hemisphere other than Antarctica to ensure they are not disproportionately impacting the skill of our reconstructions."*

**3.4 In the Discussion and Conclusions section (L373-377) the authors claim that the three listed points (which are very obvious and not really helpful) reduce the extent to which ENSO impacts their reconstructions. However, as under point 3.2, they don't show anywhere in the manuscript whether at all and if so to what extent these points affect their reconstructions. Analysing this possible impact is what would be of interested here. For this I suggest e.g. comparing reconstructions with randomly sampled proxy records with reconstructions that only include "good" records (according to points 1)-3) ). This would then allow the authors to make a statement in the direction they aim for here, but again with the caveat that the results may only be valid for the specific model in case the agreement of the model with reality is not sufficiently good.**

We agree with C.D.'s suggestion and have generated a further set of reconstructions that exclude proxies from regions we have calculated to be sensitive to ENSO. A new Figure has been added to the manuscript and is shown below. In addition, the results section has been updated with an additional paragraph which reads:

*"Removing these ENSO sensitive proxies from our SH-wide reconstructions has a small, but negative impact on the proportion of skilful reconstructions we are able to produce for both SAT and precipitation (Figure 12). Their absence also reduces the minimum skill values for the 5th percentile for all precipitation-derived reconstructions across all network sizes (Figure 12d,e,f). A smaller effect is visible for SAT-derived reconstructions calibrated with a 31 year window, but only for smaller network sizes. Given the minimal extent to which ENSO appears to modulate the proxy-SAM relationship, removing these proxies, which may otherwise enhance the regional diversity of a network, results in a net degradation of the signal to noise ratio in our reconstructions. On the other hand, reconstructions using only ENSO-sensitive proxies (not shown) also results in lower skill, although it is unclear what role ENSO plays due to the vastly reduced pool of proxies we can sample from in this scenario."*

[Figure]

Differing reconstruction skill achieved when sourcing proxies from the entire Southern Hemisphere (yellow envelopes) and the entire Southern Hemisphere, excluding proxies whose teleconnection with SAM have a significant (p < 0.05) correlation with ENSO (hatched regions in Figures 9 and 10; red envelope). The correlation between a SAT or precipitation-derived reconstruction and the SAM is on the y-axis, while the number of sites used (n = 2:70) in a reconstruction is on the x-axis. Shaded regions represent the range between the minimum of the 5 th percentile and the maximum of the 95 th percentile for each network size, across 10,000 reconstructions  described in Section 2.3). The black lines indicate the percentage of SH reconstructions (yellow envelope) that meet or exceed our skill threshold of being able to explain 50% or more of the variability in the SAM. The red line indicates the same thing, but for those reconstructions that exclude ENSO-sensitive proxies (red envelope). The dashed black line indicates the r value required to meet our skill threshold.

**Also, I wonder why the stated points should be different for seasonal data (L376-377). I think also for seasonal data it is obvious that proxy records with a significant correlation to ENSO can negatively impact the reliability of the resulting reconstructions or that proxy records with a strong enough correlation to SAM should be used.**
We have removed the comment pertaining to seasonal differences, as it was not directly relevant to the results being discussed. Furthermore, the section of the discussion in question has been re-written and now reads as follows:

" It is not unreasonable to suspect that ENSO may be contributing towards the proxy-SAM teleconnection variance. Dätwyler et al. (2020) identify a highly variable, but centennial-average r of -0.3 between austral summer ENSO and SAM reconstructions over the last millennium. Their pseudoproxy experiments using a CESM1 ensemble show significant changes in SAT during periods of large negative SAM-ENSO correlation (their Figure 4, bottom left panel). The pattern is similar to our results (Figure 9a), with regions of significant correlation over much of Antarctica and three regions in the Southern ocean centered on roughly 60 ∘ E, 150 ∘ E and 60 ∘ W. Rather than excluding proxies whose teleconnection with SAM is significantly correlated with ENSO, we can minimise ENSO's impact simply by calibrating over a longer window thus ensuring that, while ENSO may impact these proxies, the variance of their teleconnection with SAM will be small. Its greater impact at longer windows (Figure 11) is therefore minimised as the variance of the proxy-SAM teleconnection is smaller (Figure 8)."

**3.5 A further point I wonder about is how much the results relating to whether SAT or precipitation record-based reconstructions perform better (in terms of the currently used skill measure) go beyond what can simply be expected statistically. The authors select records based on their correlation with SAM over the calibration period. Then a skill measure that is solely based on the correlation of the reconstruction with SAM is used. Wouldn't the result whether SAT- or precipitation-based SAM reconstructions have higher skill then simply depend on the distribution of the correlations of SAM with SAT/precipitation in the model, that is, wouldn't the reconstructions based on the climate variable with a higher proportion of correlations (absolute values) that are**

**bigger or equal than 0.3 necessarily yield higher skill values? Or in other words, if in the model you have a higher proportion of correlation above 0.3 of SAT or precipitation with SAM, then it would be much more likely by random selection to catch pseudoproxy records that correlate strongly with SAM and hence much more likely to obtain a "skilful" reconstruction.**
**In conclusion, the results would exclusively depend on how the correlations in the model between SAT/precipitation and SAM are distributed (not spatially) and you don't even need to do reconstructions to know the outcome/answer.**

We would suggest that this is not the case, primarily due to the need to consider the spatial density of these highly correlated sites. If one field has higher proxy-SAM correlations, but they are all clustered in a geographically limited area, they do not provide much independent information to the reconstruction. This means local noise is not cancelled out and results in a degradation of the resulting reconstruction. In any case, reconstructions are not created primarily to assess the relative skill of SAT and precipitation proxies, but rather to assess the impact or calibration length and network size on reconstruction skill.

**Minor points**
**4 L7: I think it is not necessary to introduce the abbreviation SAT here (it is done on L73).**
'(SAT)' has been removed

**5 L7-9: Unclear sentence. "range in reconstruction skill": Does that range in any sense relate to non-stationarity? Where is that range obtained from? If it relates to non-stationarity the sentence does not make sense, because in the first part of it you say that non-stationarity plays a negligible role in reconstructions (where skill also belongs to).**

We agree with C.D. that this was not clear. Due to the inclusion of new results, this paragraph has been re-written and now reads:

*"Low frequency variability exists in the teleconnections between our pseudoproxies and the SAM that cannot be explained by climatic noise (stochastic variability). CM2.1 simulates, at maximum, 9% of land points as being non-stationary as defined by Gallant et al. (2013) (using precipitation as a proxy and a 91 year running window), although the odds of creating a proxy network with a high proportion of non-stationary sites remains relatively low (Figure A1). Non-stationary proxies, as defined here, do not seem to modulate SAM-proxy teleconnection strengths or impact on reconstructions greatly, as emphasised by the weak relationship between reconstruction skill and the number of non-stationary proxies in a reconstruction (Figure 7a). The exceptions are SAT-derived reconstructions with a longer calibration window (61 or 91 years), suggesting that at larger network sizes, care should be taken to minimise the proportion of non-stationary proxies. While assessing the stationarity of a proxy-SAM correlation is more difficult in the real world, we suggest that multiple methods be employed where possible such as in Dätwyler et al. (2018)."*

**6 L13: I would remove "nominally" here.**
It has been removed.

**7 Figure 1 and caption:**
**7.1 GFDL CM2.1 and ERA not defined yet (and I think it is not done anywhere else).**
**7.2 Please be consistent in how you refer to the panels (a)-(d). Here for example you use (a), (c), b), d), b and d.**
**7.3 Very minor detail: Consider switching (a) and (c) with (b) and (d) so that in the text there reference to (a) and (c) comes before (b) and (d).**
**7.4 One colour bar and one x-axis label would be sufficient I think.**
**7.5 Different font sizes on the colour bar are used (1 and 0 is significantly larger than -1, - 0.5 and 0.5). Same for the y-axes (S from 30°S on is larger than the rest). In addition, the number next to the colour bar are extremely close to it. Add some space, because one is tempted to read -0.5 on the red half of the colour bar. Please pay attention to such details!**
**7.6 Why are contours at r ≥ |0.3| of relevance here? This seems arbitrary.**
**7.7 I don't think a the reference for ERA-Interim is needed here (you provide it on L97).**

As figure 1 will be replaced in the following submission, these points do not necessitate a response (though we gratefully acknowledge C. D.'s suggestions and have made changes to the new figure where applicable).

**8 L15: "... have been linked ...": Here results from the literature are presented, but your own Figure 1 is added as reference. You should clearly separate your own results from other people's work. I think I understand the intention here (which I guess was to refer to Figure 1 as being in line with results from the literature) but this needs to be rephrased (and I'm not sure I would include your own result in the introduction here, but this is debatable).**

Due to Figure 1 being replaced in the new version of the manuscript, this reference has been removed.

**9 L22-24: Ok, but feels like a bit out of place and irrelevant here. I would remove this.**

We agree with C.D. that this did not fit here. This has been removed and the appropriate references have been added to the sentence at the end of the following paragraph. It now reads:
*"These relative contributions are important for understanding projected future changes given the impact of the SAM not only on regional weather patterns, but also large scale ocean circulation and heat uptake (Russell et al., 2006; Marini et al., 2011; Liu et al., 2018), and even the marine carbon cycle (Lovenduski et al.,* $_{30}$*2007; Lenton and Matear, 2007; Le Quéré et al., 2007; Huiskamp and Meissner, 2012; Hauck et al., 2013; Huiskamp et al.,2015; Keppler and Landschützer, 2019) and how this may change in the future."*

**10 L26-27: If I remember correctly there is also some work by Dave Thompson that might be relevant here and could possibly be cited. Please check and add if you feel like (and if you don't want to include it, it could at least be interesting to read :) ).**

We thank C.D. for the suggestion and assume they are referring to the J. Clim. paper by Li et al. in 2019? While not strictly appropriate for citing here, we appreciate being made aware of this research!

**11 L28: "as derived" sounds strange to me.**
We believe it to be grammatically appropriate in this context

**12 L29: "long-term" missing before "context"?**
We agree this would add clarity. 'a long-term' has been added before 'context'

**13 L29: Unclear. Do you mean the past five decades by "present day"? Please be specific.**
We agree. This now reads:
The sentence has been rephrased. It now reads:
*"It is important to place these observed multi-decadal trends over the last five decades into a long-term context in order to understand the contributions of forced and natural variability."*

**14 L29-30: I think the structure of this sentence does not work. I have the impression that by "present day changes" you mean the same as by "observed multi-decadal trends". Maybe just place a dot after "variability" (or otherwise rephrase).**
The sentence has been rephrased. It now reads:
*"It is important to place these observed multi-decadal trends over the last five decades into a long-term context in order to understand the contributions of forced and natural variability."*

**15 L30: "... this, this ...". Not very elegant…**
We agree "Following on from this," has been removed.

**16 L30-32: Does all this really follow from the previous sentence?? I'd remove this sentence.**

As noted above, we include this sentence to give a brief, broader context  on the importance of the SAM beyond its impacts on local, Southern Hemisphere climate . We believe this is now better integrated into the text.

**17 L35: I don't think "meanwhile" is the appropriate word here.**
We believe this is a correct use of the word

**18 L39: "can be made by examining changes" sounds strange / weird formulation…**
This has been rephrased, it now reads:

*"Paleo-reconstructions are generated by examining changes preserved in natural environmental archives (biological, chemical and physical records), that are sensitive to climatic impacts of the mode of variability being reconstructed."*

**19 L41: "... sensitive to both precipitation or surface air temperature ..." you can't use "both" and then "or".**
We thank C. D. for spotting this error, it has been corrected to:
*"In the case of the SAM, this has traditionally been achieved by finding proxies that are sensitive to precipitation or surface air temperature changes associated with the two different phases of the SAM."*

**20 L42: Why is SAT not define here but only later (L73 or so)?**
We thank C. D. for identifying this error. In fact, SAT is now defined on line 15:

*"Positive and negative phases of the SAM have been linked to changes in surface air temperature (SAT) and precipitation …"*

**21 L42: "For example ...". I feel like this sentence does not belong here. You already discussed this before.**
This has been removed.

**22 L49: Is particle dust really relevant here and for SAM? Seems to be out of place. I'd remove this.**
It has been removed.

**23 L55: Not sure about the formating here. Think I would write "... (CPS; Abram et al. ...)".**
We agree, this has been changed accordingly.

**24 L62: No comma after "regional".**
This has been removed.

**25 L64: "comparing" instead of "compare".**
In this case, both are grammatically acceptable and we prefer the latter.

**26 L74: "a significant positive correlation" or "significant positive correlations".**
We thank C.D. for spotting this error. 'correlations' has been corrected.

**27 L78-81: Isn't this true also for longer calibration windows?**
This is true, we have amended the text and now reads:
*"Finally, when considering multi-decadal calibration periods, stochastic noise or another climate signal (e.g. ENSO) can modulate the correlation strength between, for example, South American precipitation and the SAM without the precipitation record being classified as non-stationary (Yun and Timmermann, 2018)."*

**28 L82: Do you really "quantify" the "uncertainties" mentioned?**
We quantify the reconstruction uncertainties due to calibration window length, geographic distribution of proxies and non-stationarities/ climatic noise, presented within the context of our model framework.

**29 I noticed a mix of past and present tense in the methods section. I think generally only one should be chosen and then used consistently.**

We thank C.D. for spotting this. This has now been made consistent.

**30 L89: Using "is" here sounds strange ("The data ... is ...").**
We thank C.D. for spotting this error, it has been corrected to 'the data are'.

**31 L94: ERA-40 does not appear in Figure 1b and d, but the way you refer to Figure 1 it Should.**
This section has been rephrased and updated and should now be free from ambiguity. It now reads:
*"CM2.1 is selected due to its good representation of the SAM compared to similar models from the CMIP5 and CMIP6 archives (Bracegirdle et al., 2020) while Karpechko et al. (2009) find performance to be favourable when compared to ERA-40data. The spatial structure of the SAM is well simulated, accurately capturing the centre of action over the Pacific, while being slightly too zonally symmetric on the eastern half of the Southern Hemisphere (Raphael and Holland, 2006). Importantly for our purposes, CM2.1 accurately simulates the latitude at which the SAM transitions from its positive to its negative phase (as expressed via regression onto 850hPa winds) over South America, which many models of a similar age and computational complexity fail to achieve (Raphael and Holland (2006), their Figure 4b). The amplitude of the model SAM index is comparable with observations (Raphael and Holland, 2006), although its variability is larger than observed (Karpechko et al., 2009)."*

**32 L97: You miss saying that the reanalysis data are correlated with the Marshall SAM index.**
This has been addressed in the updated paragraph, seen in the previous point.

**33 L100: What do you mean by "transitions from its positive to its negative node".**
We agree that this could have been clearer. It now reads:

"Importantly for our purposes, CM2.1 accurately simulates the latitude at which the SAM transitions from its positive to its negative phase (as expressed via regression onto 850hPa winds) over South America, which many models of a similar age and computational complexity fail to achieve (Raphael and Holland (2006), their Figure 4b)."

**34 L104-105: Yes, but the present study is about annual reconstructions and later on in Section 4 it is mentioned that CMIP5 generation models have issues in representing SAM-SAT relationships on a seasonal scale. Hence, saying that Gallant et al. also use**

**CM2.1 due to its seasonal skill (austral winter) is a very incomprehensible argument for using CM2.1 (and why is "austral winter" capitalised?)**

While our aim here was to cite prior studies that had validated the use of CM2.1 for SAM research on seasonal rather than just annual-mean time-scales, we agree that its inclusion here could confuse and as such, this sentence has been removed.

**35 L110: "also"?**
We thank C. D. for spotting this error. 'Also' has been removed here.

**36 L111: "also"?**
This is correct, referring to the 'physical/chemical/biological' nature of the proxy in addition to the underlying variability of the climate which is identified in the previous sentence.

**37 L116-117: Strange sentence.**
**37.1 I think it should read "By using a model framework ...".**
**37.2 "robustly". Really?**
**37.3 "windows in time" → "time windows".**
**37.4 "the skill of index reconstruction" sounds weird.**
This sentence has been removed from the methods.

**38 L107-127: This text seems not to belong to the "Methods" section.**

This text has largely been removed, with aspects moved to the Introduction.

**39 L120: I don't think "Alternatively" is the right word here.**
It is a correct use of the word

**40 L122: What do "dust particle records" do here? I'd remove this sentence (and also it seems strange to start it with "Finally" and then you begin the next sentence with "In addition").**

We agree with C.D. and the suggested changes have been made.

**41 L123: "In addition to this, ..." "this" is very unspecific here. Please rephrase. Also, I've notice you use "this" at various places in a similar unspecific way. Please check and clarify where necessary.**

With the removal of the previous sentence, "this" is now clearly referring to the seasonal sensitivity of paleo proxies. With this in mind, the manuscript has been revised for instances where 'this' is used in an unambiguous manner.

**42 L125: "... during an entirely different season." instead of "... during a different season entirely."?**

Both formulations are grammatically correct. We have left the sentence as is.

**43 L127: Consistency with what? Why should annual means be more consistent than seasonal analyses? I'm not convinced by this argument.**

We agree with C.D. and have removed the reference to 'consistency'. This sentence now reads:

*"For the sake of simplicity, we simply employ annual mean (Jan-Dec) fields for sea level pressure,surface air temperature and precipitation and focus instead on the impact of network size and calibration window length rather than seasonal effects."*

**44 L130: As far as I know, most commonly latitudes between 40°S and 65°S (rather than 60°S) are used for the definition of SAM and only a minority of studies use a different definition. Is there any reason that would speak for 60°S instead of 65°S?**

C.D. is correct that 65S is the most common latitude selected for the definition of the SAM. Our use of 60S was following the method of Gallant et al. 2013, and along with other methods such as using the first EOF of sea level pressure, we believe it to be an acceptable definition and one that should not meaningfully alter the results presented.

**45 L134: I don't think "established" is suitable here. Maybe "modelled" or something in that direction? Also I'd use plural ("running correlations").**
This has been changed as suggested.

**46 L132 (and whole manuscript): Be concise and consistent throughout the manuscript when using "proxy", "proxy record", "proxy data" and "proxy archive". Here I would maybe use "proxy record".**
While these are synonymous terms, we have changed the manuscript to use a more limited and consistent number of terms. Namely "proxy/ies" and "proxy record(s)".

**47 L132 (and whole manuscript): I just noticed here that you use "the SAM" and three lines later only "SAM". Please be consistent.**

We thank C.D. for spotting the missing 'the'. This was an error and has been fixed. The rest of the manuscript has been likewise amended.

**48 L135 (and whole manuscript): Here you write "non-stationary" and on the next line "nonstationary". Please be consistent.**

We thank C.D. for identifying this inconsistency. We have revised the manuscript so that all instances are written as 'non-stationary'.

**49 L147: I'd replace "-" with a comma and on the next line there is a "the" missing before**

**"SAM index".**

The '-' has been replaced with a comma as suggested and 'the' has been inserted before 'SAM index'.

**50 L148-149: I don't think this sentence is correct. You say the red noise is a combination of random Gaussian noise and lag-1 autocorrelation of a climate variable time-series. But random Gaussian noise is a time-series, whereas the autocorrelation of a time-series is a number. The formula is correct, but the error happened in the attempt to put it into words I think.**

We agree that this could have been articulated more clearly. It has been rephrased in text and how reads:
*"The red noise is a combination of random Gaussian noise ($\eta_v(t)$) and autocorrelation ($\beta$) of the SAT or precipitation time-series at a lag of one year multiplied by the Gaussian noise ($\beta\eta_v(t-1)$)."*

**51 L152-154 sounds a bit strange / weird formulation. Please rephrase.**
It has been re-phrased. It now reads:

*"Therefore, if the time-series from our model proxy has a running correlation that falls outside the confidence interval,we consider that proxy to be non-stationary with the SAM in that temporal window, as it is unlikely to be affected by stochastic processes alone."*

**52 L165: "metric"? Do you mean "climate variable". Also "deemed" sounds strange to me.**
Metric has been changed to 'climate variable' as requested. 'Deemed' is a correct use of the word here.

**53 L169: "For this reason" does not sound logic here.**
We agree, this sentence has been removed.

**54 L182: Strange sentence. Is there a word missing?**
We thank C.D. for spotting this error. It should have (and now does) read:
*"... for each network size."*

**55 L185-188: I'm happy with the choice of CPS as reconstruction method, but I think you can't say in such a general sense that it is considered to be superior to other methods.**

We agree, this statement has been removed.

**56 L192: Where do you define how your model "nino3.4" index is calculated. I think this should be mentioned somewhere.**

We agree. The definition of the Nino 3.4 index (and by extension, our method for calculating it) has been added to the methods section. It reads:

*"Running correlations of SAT/precip and the SAM are correlated with the model Nino3.4 (n3.4) index to investigate the role the El Nino Southern Oscillation (ENSO) may play in modulating the pseudoproxy-SAM teleconnection. The model n3.4 index is calculated as the sea surface temperature anomaly in the region bounded by $5^oN$ to $5^oS$ and $170^oW$ to $120^oW$."*

**57 L194: Where do you introduce the abbreviation "n3.4"?**

It is now defined in the paragraph referenced in the previous comment.

**58 At this point I will stop pointing to typos, inconsistencies and unclear sentences, because, as explained above, I think major revisions of the figures and text is necessary anyway.**

**59 Maybe a single last comment to Figure 3, its caption and the use of "teleconnection":**
**- Do you really to display the legend four times?**
We have removed three of the legends as requested.
**- The sentence that starts with "This is therefore ..." does not follow from the previous.**
This sentence has been removed.
**Also I don't understand why you say "running" windows.**
We thank the C. D. for identifying this error. "Running" has been replaced with "calibration" in this sentence.
**- Sometimes you use "teleconnection" as substitute for "(running) correlation". These are generally not necessarily the same. While teleconnections are more "general", correlations may only capture a specific aspect of a teleconnection.**
We have fixed this where appropriate.

---

## Referee Report (RR1)

**Review of Huiskamp and McGregor**

**1) General comment:**

I appreciate that most of the comments from my previous report have been addressed by the authors. The new version reads well, and the methodology is easier to follow. The authors often explain well to what extent their study can be useful for "real-world" SAM reconstructions and what are the limitations, which is very appreciated. I agree for the publication of this article that would be a very nice support for other pseudo-proxy experiments and real-world reconstruction of SAM variability. Although the Results section is clear and interesting, I still have concerns about a few technical aspects. There is notably something wrong, or unclear at best, with the non-stationarity test based on the one from Gallant et al. (2013).

I sincerely apologize for the time delay of my second report.

**2) Revisions/Comments:**

**L.124:** "*For the sake of simplicity, we employ annual mean (Jan-Dec) fields for sea level pressure, surface air temperature and precipitation and focus instead on the impact of network size and calibration window length rather than seasonal effects.*"

I see this statement has been added to address a comment from the other reviewer (C. D.). Even though the authors were already using annual timeseries in the previous version, there are actually issues with this. The proxy network size and the calibration window length recommended at the end of the study could be affected by the fact of adding seasonal effects in proxy records. Indeed, one of the main purpose of this study is **L. 80:** "*2) How does the geographical distribution of the proxies affect reconstruction skill?*", but the effect of geographical distribution of proxies related to SAM variations is also dependent of the season targeted by these proxies (as stated by the authors in the same paragraph). Also, **L. 81:** "*3) Are any regions in our model framework prone to producing non-stationary proxies and what could be modulating the SAM-proxy teleconnection?*". Would these non-stationarities be still present for the seasonal averages of each region? How can we be sure that the non-stationarities detected in this study could not come from the correlations altered by the use of annual averages?

If this study aims at better understanding how to reconstruct the SAM in the real world, what can the reader learn from the output of this model-based study if it makes conclusions from an unplausible situation in the "real world" (*i.e.* all proxies measuring annual averages of climate)? The authors should try to add further discussions to argue this choice because the sake of simplicity is not enough since reconstructing the SAM in the real world is a complex problem.

**L. 150:** "*It should be noted that as a 95% confidence interval is used, non-stationarity will be falsely identified 5% of the time, hence we define a grid point as non-stationary only if the running correlation falls out of the confidence interval more than 10% of the time, or 50 of the 500 years; more than double the 5% we might expect by chance alone.*"

According to the author's scripts, maps from Fig. 6 are drawn at each 0.1 step for the number of times the running correlations falls out of the ones drawn by Monte-Carlo repetitions. So,

if I well understand, this means that it is not 50 of the 500 years but rather 10% of the 471, 441 and 411 sliding time frames tested for the 91, 61 and 31 length cases (respectively). If I am right, the authors should avoid saying this is equivalent to 50 of the 500 years, same as for the Fig. 6 caption. More generally, saying that a given year out of 500 has a running correlation doesn't make sense, unless if taken as a centre of a time frame, but there are not 500 of them here.

**L. 154:** "*As correlations are bounded between +/-1, the running correlations are converted to Fisher Z-scores:*"

Yes, correlations are bounded by +/-1, so? I don't see why this is stated here while not a single Z-score is discussed later. Are they used to compute significance levels within the Monte-Carlo framework? If yes, it should be said somewhere because in the present form, it feels these Z-scores come from nowhere.

**L. 192:** This is not clear why a band-pass filter is applied to n3.4. How has the latter been chosen? Would this significantly change the correlations significances calculated by the authors if not using the band-pass filtering?

**L. 188-196:** Some uses of "correlated" are a bit confusing in those lines since it is describing how to determine which proxy/SAM teleconnections are effectively correlated with ENSO variations. It would be clearer for the reader to simply say that a correlation coefficient is computed instead.

**L. 250:** The paradigm of the choice between the calibration window length and the amount of available proxy data fully covering this time frame is very well discussed here. It is very challenging and strongly affects reconstructions when working with real world data. However, I would have thought that a reconstruction method such as the CPS is not so much affected by this problem because correlations (weights) can be computed for each individual overlap periods of the proxies and the SAM (which can then be maximised for each). This is different for methods like the PCR (principal component regression) for which it is impossible to diagonalise the proxy matrix if it has missing data, which thus makes important the choice of the length of the calibration period common to all proxies.

**L. 275:** Similarly to my comment for **L. 150**, there is something wrong here. If the authors are effectively considering running correlations for 31-, 61- and 91-time length, falling out of the 95% range in more than 10% of the time would means 48 times out of 471 for 31-year windows (because there are 471-time windows of size 31 in a 500 year-long period). In the same way, falling out of the 95% range in more than 10% of the time would means 45 times out of 441 for 61-year windows and 42 times out of 411 for the 91 ones. In Fig. 6 it is apparent that the authors use contours at level 50 (*i.e.*, when colours turn to orange), and not at the true 10% levels, specific to each time length used to compute the running correlations (see above). This means that the authors are not rejecting stationarity when 10% of running correlations falls out of the 95% Monte-Carlo range. They are actually doing it for ~10.6% (50/471) for 31-year time length, the ~11.3% (50/441) for 61-year time length, and ~12.2% for 91-year time length. The authors need to find a way to fix Fig. 6 and subsequent analyses

if they want to keep the rejection of stationarity at a 10% level. Otherwise, they should adapt the main text to this fact.

**L. 277:** *"For SAT, 4% (31 year window) and 8% (61 and 91 year window) of land cells are non-stationary, while for precipitation 6% (31 year window, 10% (61 year window), and 9% are (91 year window)."*
These values might not be the same after addressing my last comment. There is also a missing right bracket in this sentence.

**L. 277: "***[…] despite ENSO potentially being responsible for 50% of this variance."*
50% when ENSO is band-passed filtered?  How has it been calculated?

**L. 433:** Remove "TEXT".

---

## Author Response (AR2)

**Response to reviewers (second round) for manuscript cp-2020-133**
**Willem Huiskamp and Shayne McGregor**
**23/07/21**

We thank both the anonymous reviewer and the editor for their helpful comments and suggestions. Below, we address each comment with the reviewers text in bold, and our responses following. Excerpts from the text are presented in italics, with changes in red.

**Reviewer 1:**
**L.124: " For the sake of simplicity, we employ annual mean (Jan-Dec) fields for sea level pressure, surface air temperature and precipitation and focus instead on the impact of network size and calibration window length rather than seasonal effects."**

**I see this statement has been added to address a comment from the other reviewer (C. D.). Even though the authors were already using annual timeseries in the previous version, there are actually issues with this. The proxy network size and the calibration window length recommended at the end of the study could be affected by the fact of adding seasonal effects in proxy records. Indeed, one of the main purpose of this study is L. 80: "2) How does the geographical distribution of the proxies affect reconstruction skill?", but the effect of geographical distribution of proxies related to SAM variations is also dependent of the season targeted by these proxies (as stated by the authors in the same paragraph). Also, L. 81: "3) Are any regions in our model framework prone to producing non-stationary proxies and what could be modulating the SAM-proxy teleconnection?". Would these non-stationarities be still present for the seasonal averages of each region? How can we be sure that the non-stationarities detected in this study could not come from the correlations altered by the use of annual averages? If this study aims at better understanding how to reconstruct the SAM in the real world, what can the reader learn from the output of this model-based study if it makes conclusions from an unplausible situation in the "real world" (i.e. all proxies measuring annual averages of climate)? The authors should try to add further discussions to argue this choice because the sake of simplicity is not enough since reconstructing the SAM in the real world is a complex Problem.**

We appreciate the concern raised by the reviewer regarding our selection of annual-mean climatological fields for our pseudoproxies and its relevance to real-world reconstructions. Our motivation was as follows: 1) As stated on L124, limiting our experimental parameter space and avoiding not only the issue of seasonality, but the problems that arise with this (i.e. ice cores are annually resolved - should we be combining these with seasonal pseudo-tree rings?). 2) As noted in the discussion (L405), current state of the art climate models are considerably less skillful in simulating the SAM on a seasonal time-scale than in the annual-mean. The use of seasonal data would be less valuable if the location/ intensity of the westerlies show greater biases. 3) (L405) The hope that reconstructing the SAM on an annual-mean time-scale would smooth out high-frequency noise and enhance the signal to noise ratio of our reconstructions.

This should appear in the methods rather than the discussion, and as such has been moved. The updated methods read as follows:

*"With this in mind, we employ annual mean (Jan-Dec) fields for sea level pressure, surface air temperature and precipitation for the following reasons. 1) The CMIP5 generation of models (including CM2.1) are less skilful at representing seasonal variability in the SAM-SAT relationship thanover the annual-mean (Marshall and Bracegirdle, 2015); 2) Reconstructing the SAM on an annual-mean time-scale should smooth out high-frequency noise in the proxies and enhance the signal to noise ratio of our reconstructions. 3) To simplify the experimental parameter space and focus instead on the impact of network size and calibration window length rather than seasonal effects."*

The reviewer's second concern is regarding the validity of these results, i.e. - given the enormous complexity of paleo-SAM reconstructions and the nuances involved in calibrating individual proxies, what value does a simplified framework such as this provide? To begin with, our method finds analogues in Abram et al. 2014 and Dätwyler et al. 2018, both of whom reconstruct an annual-mean SAM despite the inclusion of highly seasonal proxies such as tree rings (as well as annually resolved proxies such as isotopes derived from ice cores), so we disagree that this represents an entirely 'unplausible(sic) situation in the real world'. We are simply being consistent by averaging all our data annually.

Secondly, the impact of this approach as opposed to a purely seasonal approach may be hinted at in the results of Dätwyler et al. 2020, which we discuss on lines 413-417. This shows, for reconstruction skill at least, values that are similar to those we find in our results.

**L. 150: "It should be noted that as a 95% confidence interval is used, non-stationarity will be falsely identified 5% of the time, hence we define a grid point as non-stationary only if the running correlation falls out of the confidence interval more than 10% of the time, or 50 of the 500 years; more than double the 5% we might expect by chance alone."**

**According to the author's scripts, maps from Fig. 6 are drawn at each 0.1 step for the number of times the running correlations falls out of the ones drawn by Monte-Carlo repetitions. So, if I well understand, this means that it is not 50 of the 500 years but rather 10% of the 471, 441 and 411 sliding time frames tested for the 91, 61 and 31 length cases (respectively). If I am right, the authors should avoid saying this is equivalent to 50 of the 500 years, same as for the Fig. 6 caption. More generally, saying that a given year out of 500 has a running correlation doesn't make sense, unless if taken as a centre of a time frame, but there are not 500 of them here.**

We thank the reviewer for spotting this error! They are correct that, rather than a time-frame of 50 years, the 10% threshold should be 47, 44 and 41 years for the 31, 61 and 91 running correlation windows, respectively. As such, Figure 6 in the manuscript has been updated and the caption now reads as follows:

*"Figure 6. Number of years at each grid point where the 31 year (a,b), 61 year (c,d) and 91 year (e,f) running correlation between SAT (a,c,e) or precipitation (b,d,f) and the model SAM falls outside the 95% 'stationarity' confidence interval (Section 2.2). As per our definition of non-stationarity, regions which fall outside this interval 10% of the time or more (≥47, 44 and 41 years for the 31, 61 and 91 windows, respectively) are highlighted with solid black contours and are considered to be non-stationary"*

**L. 154: "As correlations are bounded between +/-1, the running correlations are converted to Fisher Z-scores:"**
**Yes, correlations are bounded by +/-1, so? I don't see why this is stated here while not a single Z-score is discussed later. Are they used to compute significance levels within the Monte-Carlo framework? If yes, it should be said somewhere because in the present form, it feels these Z-scores come from nowhere.**

We thank the reviewer for noticing this - it was not written clearly enough. They are indeed correct that the Z-scores are used to calculate the significance levels within the Monte-Carlo framework. This sentence has been updated and now reads as follows:

*"The running correlations are converted to Fisher Z-scores to ensure they are normally distributed for the calculation of confidence intervals"*

**L. 192: This is not clear why a band-pass filter is applied to n3.4. How has the latter been chosen? Would this significantly change the correlations significances calculated by the authors if not using the band-pass filtering?**

The band-pass filter is applied to the model nino3.4 index due to its exclusive comparison with time-averaged values of proxy-SAM correlations. By calculating these running means between (for example) precipitation and the model SAM, we are smoothing out variability shorter than 31, 61 or 91 years. To ensure the power spectrum of the model ENSO record was consistent, they were band pass filtered before being correlated with the SAM-proxy running correlation records. We would expect these correlations to change if we had not performed the smoothing, but the results would be spurious.

The choice of the Nino3.4 index is due to its ability to optimally represent the character and evolution of El Nino and La Nina events (Bamston et al., 1997; Trenberth and Stepaniak 2001). These references have been added to the methods section.

**L. 188-196: Some uses of "correlated" are a bit confusing in those lines since it is describing how to determine which proxy/SAM teleconnections are effectively correlated with ENSO variations. It would be clearer for the reader to simply say that a correlation coefficient is computed instead.**

We agree that this could have been more clearly articulated. As a result, this sentence has been replaced and now reads as follows:

*"To investigate the role ENSO may play in modulating the pseudoproxy-SAM teleconnection, a correlation coefficient is calculated between running correlation time-series' of SAM-SAT/precipitation and the model Nino3.4 (n3.4) index at each grid point."*

**L. 250: The paradigm of the choice between the calibration window length and the amount of available proxy data fully covering this time frame is very well discussed here. It is very challenging and strongly affects reconstructions when working with real world data. However, I would have thought that a reconstruction method such as the CPS is not so much affected by this problem because correlations (weights) can be computed for each individual overlap periods of the proxies and the SAM (which can then be maximised for each). This is different for methods like the PCR (principal component regression) for which it is impossible to diagonalise the proxy matrix if it has missing data, which thus makes important the choice of the length of the calibration period common to all proxies.**

The reviewer is correct in their observation that a weighted CPS does not explicitly require the calibration period to be uniform across all proxy records, unlike methods such as PCR. It does however complicate the weighting of these proxies consistently. For example, a proxy calibrated over 91 years will likely have a lower correlation coefficient than another proxy calibrated over 31 years resulting in a reduced weighting in the final reconstruction, despite being better constrained. There is clearly no ideal solution here, but in this instance, we believe being consistent in our application of calibration window length at least eliminates this added complexity.

**L. 275: Similarly to my comment for L. 150, there is something wrong here. If the authors are effectively considering running correlations for 31-, 61- and 91-time length, falling out of the 95% range in more than 10% of the time would means 48 times out of 471 for 31-year windows (because there are 471-time windows of size 31 in a 500 year-long period). In the same way, falling out of the 95% range in more than 10% of the time would means 45 times out of 441 for 61-year windows and 42 times out of 411 for the 91 ones. In Fig. 6 it is apparent that the authors use contours at level 50 (i.e., when colours turn to orange), and not at the true 10% levels, specific to each time length used to compute the running correlations (see above). This means that the authors are not rejecting stationarity when 10% of running correlations falls out of the 95% Monte-Carlo range. They are actually doing it for ~10.6% (50/471) for 31-year time length, the ~11.3% (50/441) for 61-year time length, and ~12.2% for 91-year time length. The authors need to find a way to fix Fig. 6 and subsequent analyses if they want to keep the rejection of stationarity at a 10% level. Otherwise, they should adapt the main text to this fact.**

As with the previous comment for L154, these percentages have now been calculated, so that each accurately represents a true 10% of the total time interval.

**L. 277: "For SAT, 4% (31 year window) and 8% (61 and 91 year window) of land cells are**

**non-stationary, while for precipitation 6% (31 year window, 10% (61 year window), and 9% are (91 year window).”** These values might not be the same after addressing my last comment. There is also a missing right bracket in this sentence.

The reviewer is correct. These percentages have been recalculated and the changes read as follows:
*“For SAT, 6% (31 year window) and 11% (61 and 91 year windows) of land cells are non-stationary, while for precipitation 7% (31 year window) and 14% are (61 and 91 year windows)”*

**L. 277: “[…] despite ENSO potentially being responsible for 50% of this variance.” 50% when ENSO is band-passed filtered? How has it been calculated?**

As is stated in the methods, all calculations pertaining to the model ENSO index are those that have been band-pass filtered. We agree that the 50% figure was not clearly attributed to regression analysis displayed in Figure 11. A new reference has been included in this sentence as well as a slight re-wording. It now reads:

“*This is of lesser consequence, however, as most sites show little variance in SAM-SAT teleconnection at this longer window length (Figure 8c; most regions have an rstd<0.1) despite ENSO potentially being responsible for 50% or more of this variance (Figure 11c)*”

**L. 433: Remove “TEXT”.**

This has been removed.

**Editor's comments**

**Title of section 1.1. Is it reconstructions of 'SAM variance' or of 'SAM variations' ? 'variance' may be misleading because of the association with the statistical definition of the variance.**

We agree with the editor that this could be clearer. As such, we have changed the title from 'variance' to 'variations'.

**Lines 83-90. Maybe add some references on the interest and limitations of pseudo-proxies?**

We thank the editor for the suggestion. We have included references to the papers by Mann and Rutherford (2002) and Mann et al. (2007), which introduces the concept of pseudoproxies and explores the robustness of proxy-based reconstructions of climatic fields, respectively.

**Line 85. I would use 'mask' instead of 'degenerate' and I must admit that it was not easy for me to make the link between the first half of the sentence that mentions 'non-climatic**

**noise' and the second half focused on a 'real' change in the strength of teleconnection due to variability in the climate. I think I finally got the message but I had to read several times the sentence so maybe a rephrasing would be helpful**

We agree that this was not sufficiently clear. The sentence has been split and rephrased, and is hopefully now unambiguous. The new excerpt reads as follows:

*"These 'perfect' proxies are free from non-climatic noise that may degrade a teleconnection signal between a real proxy and the SAM. Instead, our pseudoproxies isolate changes in teleconnection strength due to underlying variability in the climate only."*

**Line 108, it is not clear what is referred to in 'in this instance'. Is it Karpechko et al. (2009)? Another issue for Figure 1 is the comparison of the results of control simulations with the observations over past decades that likely includes a forced trend. It is mentioned elsewhere but I think the impact of this point should be included in the discussion of the figure 1.**

We have amended this sentence to add clarity and include the editors suggestion with regards to the comparison of a time period over which a trend is observed, with a pre-industrial, stable control simulation. The new sentence reads as follows:

"*As previously noted, we should be cautious directly comparing observations spanning a brief time period (in this instance, ERA-Interim (Dee et al., 2011) data correlated with the Marshall SAM index (Marshall, 2003) over the 36 year period from 1979-2014) with a well observed SAM trend, with our model data which represents a stable pre-industrial climate spanning 500 years*"

**Line 168. If I understand well, the proxies are selected if they meet the criteria on one of the windows, not all of them, and you are only using one time window for all the proxies at the same time for each reconstruction. Maybe it would help to state this explicitly. The wording 'For each grid point of the model' may suggest that the procedure is different for each point.**

The editor has indeed interpreted our methodology correctly. We agree that this section could have been more clearly structured. Three paragraphs in total have been altered and we have explicitly included the following sentence to address the issue of calibration over one or more windows:

"*Similarly, all sites in a network are selected based upon correlations over a single window, and may therefore be absent from networks calibrated using a different window.*"

**Line 215 (and also 253-260). The larger skill when using precipitation record is interesting. Is it possible to make a link with the correlation between those records (precipitation and temperature) with SAM, as for instance illustrated in Figure 2.**

We link the correlation between these records and SAM and reconstruction skill with the inclusion of Figure 8, which depicts that precipitation sites that meet our selection criteria, while fewer in number, show reduced variation in correlation with SAM over our 10 calibration windows, when compared with SAT. This is briefly addressed in the discussion (L365). This may however be one of many contributing factors (as was discussed with C.D. in the previous review round) including the distribution of sites selected for a network and how effective this is at cancelling out regional noise (this may be different for precipitation and SAT). We leave such analyses for future work.

**Figure 10. I may have missed it but the reference to figure 10 occurs after the one to figure 11 in the text.**

We thank the editor for spotting this error. The two figures have swapped places to be consistent with when they are cited.

**Line 355. 'less exhibiting less spread', maybe suppress one 'less'.**

This has been corrected.

**Line 368 My visual interpretation is that the difference in RMSE is small between precipitation and temperature derived reconstructions. This difference should be quantified here to confirm that it 'again perform better'.**

The editor is indeed correct that the median RMSE measure shows a far smaller difference in reconstruction skill between SAT and precipitation. That being said, precipitation-derived reconstructions still perform better overall. We have amended this sentence to reflect this, it now reads:

*"If we consider median root mean square error (RMSE), precipitation-derived reconstructions perform better overall (minimum RMSE of 0.91 for SAT and 0.90 for precipitation; Figure A2). As with our threshold skill score, the RMSE shows skill is maximised by utilising a large proxy network and a longer calibration window of 61 or 91 years, though the difference in skill between SAT and precipitation is smaller."*